# Antiferromagnetic magnonic charge current generation via ultrafast optical excitation

Lin Huang[1,5], Liyang Liao[1,2,5], Hongsong Qiu [3,5], Xianzhe Chen[4], Hua Bai[1], Lei Han[1], Yongjian Zhou[1], Yichen Su[1], Zhiyuan Zhou[1], Feng Pan[1], Biaobing Jin [3] ✉ & Cheng Song [1] ✉

Néel spin-orbit torque allows a charge current pulse to efficiently manipulate the Néel vector in antiferromagnets, which offers a unique opportunity for ultrahigh density information storage with high speed. However, the reciprocal process of Néel spin-orbit torque, the generation of ultrafast charge current in antiferromagnets has not been demonstrated. Here, we show the experimental observation of charge current generation in antiferromagnetic metallic $Mn_2Au$ thin films using ultrafast optical excitation. The ultrafast laser pulse excites antiferromagnetic magnons, resulting in instantaneous non-equilibrium spin polarization at the antiferromagnetic spin sublattices with broken spatial symmetry. Then the charge current is generated directly via spin-orbit fields at the two sublattices, which is termed as the reciprocal phenomenon of Néel spin-orbit torque, and the associated THz emission can be detected at room temperature. Besides the fundamental significance on the Onsager reciprocity, the observed magnonic charge current generation in antiferromagnet would advance the development of antiferromagnetic THz emitter.

Antiferromagnet (AFM) with the intrinsic frequency of terahertz (THz) and high stability against disturbing external fields, are prime candidates to develop new types of memory and logic devices[1–3]. In addition to the effects associated with spin-orbit torques (SOT) from the heavy metal (HM) layer[4–7], the thermo-magnetoelastic effect[7,8] is also one of the mechanisms for Néel vector switching in the AFM/HM heterostructure. In turn, the high intrinsic frequency of AFM offers an opportunity to pump ultrafast spin current and generate the ultrafast charge current. For instance, in the AFM/HM bilayer, sub-terahertz spin pumping was observed by detecting the voltage signal converted via inverse spin Hall effect (ISHE)[9,10]. Based on this spin-pumping effect, an antiferromagnetic THz emitter has been achieved, and an ultrafast charge current can be generated via ISHE[11,12]. However, whether the charge current can be directly generated in AFM single film is still unclear[13].

A direct conversion from magnons to charge current via spin-orbit coupling has been observed in ferromagnetic (Ga, Mn) As thin films. In this case, the magnetization precession can be directly converted into a charge current without additional HM layers[14]. In AFMs with local inversion asymmetry, such as CuMnAs and $Mn_2Au$[15], magnetization switching can be induced without the assistance of HM and achieved only in single AFM film[13,16,17]. Due to the local inversion symmetry breaking, a current induces non-equilibrium spin polarization. Opposite inversion symmetry breaking in the two sublattices results in staggered spin polarization, leading to the coherent switching of the moments at two antiferromagnetic sublattices. Such current-induced torque for Néel vector switching is known as the Néel spin-orbit torque (NSOT)[13,16–18]. Accordingly, in the reciprocal process of NSOT, magnonic charge current is expected to occur in AFM with local inversion symmetry breaking, when staggered non-equilibrium spin

[1]Key Laboratory of Advanced Materials (MOE), School of Materials Science and Engineering, Tsinghua University, Beijing, China. [2]Institute for Solid State Physics, University of Tokyo, Kashiwa, Japan. [3]Research Institute of Superconductor Electronics (RISE), School of Electronic Science and Engineering, Nanjing University, Nanjing, China. [4]Department of Materials Science and Engineering, University of California, Berkeley, CA, USA. [5]These authors contributed equally: Lin Huang, Liyang Liao, Hongsong Qiu. ✉e-mail: bbjin@nju.edu.cn; songcheng@mail.tsinghua.edu.cn

polarization is excited by antiferromagnetic magnetization dynamic, i.e., a fluctuation of the Néel vector **n**.

Since the frequency of antiferromagnetic magnetization dynamic is in THz region, it is extremely difficult to excite AFM magnons using on-chip waveguides as in the ferromagnetic structures. In this regard, ultrafast optical excitation is suggested as the most promising tool to explore the antiferromagnetic magnetization dynamic[19–24], and is able to excite a fluctuation of the Néel vector **n** in picosecond time scale[21]. The charge current generated from the fluctuation of **n** is accompanied by the emission of THz wave. In this work, we experimentally demonstrate the THz waveforms generated by the magnonic charge current in AFM metallic $Mn_2Au$ thin films, which are excited by ultrafast optical pulse.

The crystalline structure of $Mn_2Au$ is shown in Fig. 1a, whose magnetic sublattices exhibit a local inversion symmetry breaking. The femtosecond laser pulse triggers the orientation fluctuation[12,25,26] and magnitude reduction[27] of the local magnetic moments, resulting in locally non-equilibrium spin polarization ($\sigma_A = -\sigma_B$) with opposite signs on the Mn sublattices. This staggered spin polarization of the electrons is directly converted into a charge current by spin-orbit coupling due to the local inversion symmetry breaking (see Supporting Information Note1). The magnonic charge current can be expressed as $\mathbf{J} \propto \boldsymbol{\sigma}_{A,B}\hat{\mathbf{z}}$, where the charge current $\mathbf{J}$ and spin polarization $\boldsymbol{\sigma}_{A,B}$ are orthogonal (Fig. 1b, c). The observed AFM magnonic charge current generated from the magnetic moment fluctuation can be a favorable building block for antiferromagnetic THz emitter and provides a promising platform to deepen the understanding of NSOT from the view of Onsager reciprocity.

## Results

### THz signal from $Mn_2Au$ single layer

We demonstrate the AFM magnonic charge current generation in 15 nm $Mn_2Au$ thin films utilizing the THz emission spectroscopy technique. The schematic of THz emission spectroscopy setup is shown in Fig. 2a, and the coordination system ($xyz$) is defined for the laboratory frame. Ultrafast laser pulses with $y$-axis polarization are used as a pump that propagates along the $z$-axis, and the sample is arranged perpendicular to the $z$-axis with an in-plane rotation angle referred to as $\theta$ (See Supporting Information Note2). The $y$-component of the THz electric field is measured in the time domain through electro-optical sampling. The $Mn_2Au$ films are (103)-oriented and the uniaxial magnetic anisotropy (UMA) of $Mn_2Au$ films is ascribed to the uniaxial strain from the $Pb(Mg_{1/3}Nb_{2/3})_{0.7}Ti_{0.3}O_3$ (PMN-PT) (110) substrate[13,28].

A typical waveform of the THz emission is observed in PMN-PT/$Mn_2Au$ thin film (Fig. 2b), which is linearly polarized along $y$-axis and is pumped with the laser fluence of 12 uJ mm$^{-2}$. The corresponding Fourier spectrum of the THz waveform is plotted in Fig. 2c, where the

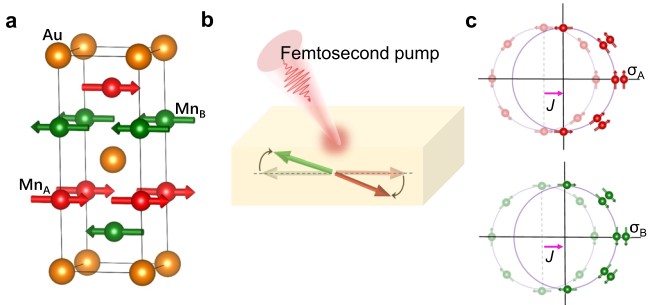

**Fig. 1 | Principle of magnonic charge current generation in $Mn_2Au$. a** The crystalline structure of $Mn_2Au$. Magnetic sublattices $Mn_A$ and $Mn_B$ are marked in red and green, respectively. **b** The orientation fluctuation, and magnitude reduction of the local magnetic moments in $Mn_2Au$ is triggered by the femtosecond laser pulse, which results in a non-equilibrium spin polarization with the opposite sign on the adjacent Mn sublattices. **c** Staggered spin polarization generated at magnetic sublattices, giving rise to the charge current by the reciprocal relationship of Néel spin-orbit torque (NSOT).

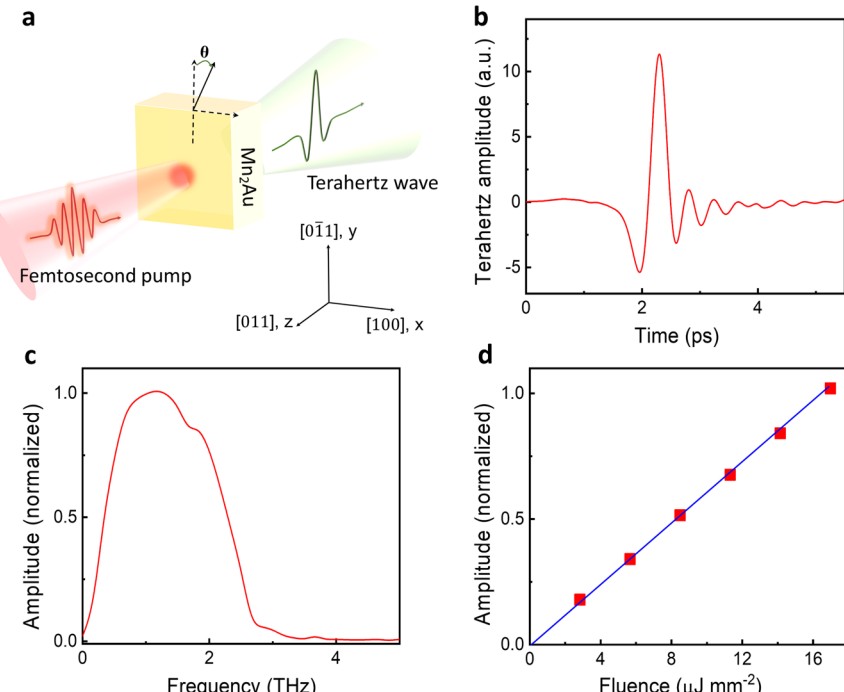

**Fig. 2 | Experimental setup and terahertz (THz) spectrum. a** The schematic of THz emission spectroscopy. The pump laser propagates along the $z$-axis and sample is set in the $x$-$y$ plane and the azimuth is denoted by $\theta$. **b** Temporal THz waveform of the THz emission from 15 nm $Mn_2Au$ single layer. **c** Fourier-transformed spectrum of **b**. **d** THz amplitude as a function of the laser fluence. The solid line is a linear regression fit to the experimental data. Both **c** and **d** spectra are normalized to peak amplitude.

effective spectral width is about 2.8 THz. The linear dependence of the THz amplitude on the laser fluence is shown in Fig. 2d and the THz amplitude does not reach saturation within the energy range.

## THz emission mechanism for single Mn₂Au layers

A comparison of THz emission between Mn$_2$Au and other AFM-based structures is shown in Fig. 3a. The results indicate that there is no spin-to-charge current conversion in the heavy metal layer Pt to enhance the THz signal of Mn$_2$Au thin film, and the behavior is significantly different from that of NiO and NiO/Pt[11]. It shows that the amplitude of THz signal from the Mn$_2$Au single layer is much higher than that of the Mn$_2$Au/Pt bilayer because the THz absorption of heavy metal layer decreases the terahertz signal (see Supporting Information Note3). Moreover, FeRh is a particularly interesting system that displays a collinear antiferromagnetic phase without inversion symmetry breaking. Thus, a 15 nm FeRh thin film[29] is used as a control sample for Mn$_2$Au, and a very weak THz signal from FeRh is observed, which is 50 times weaker than that of the Mn$_2$Au single layer (inset of Fig. 3a). The charge current generated in the opposite magnetic sublattice should

be cancel out due to the antisymmetric structure of FeRh, so that the magnetic dipole radiation in FeRh is negligible[30,31]. While, a non-vanished charge is generated in Mn$_2$Au due to the local broken inversion symmetry, in which the amplitude of the THz signal is much higher than that of the FeRh. It suggests that the mechanism of the magnetic dipole radiation[32] as THz wave emission from Mn$_2$Au thin films can be excluded. AFM Néel vector fluctuation triggered by femtosecond laser pulse results in a non-equilibrium spin polarization at two magnetic sublattices, where the staggered polarized spins convert into a charge current via the reciprocal phenomenon of NSOT. There is no obvious influence on the THz emitted signal when a magnetic field (around 4 kOe) is applied to Mn$_2$Au (Fig. 3b), indicating that the THz emission caused by antiferromagnetic order cannot be changed by the external magnetic field. This feature also reflects the advantage of antiferromagnetic THz emitter.

## Sample azimuth dependence after Néel vector switching

To clarify the relationship between the polarization direction of emitted THz signal and antiferromagnetic Néel vector **n**, we employ ferroelastic strain from the ferroelectric material PMN-PT to switch the Mn$_2$Au Néel vector[13,33]. The experimental results of emitted THz signals are summarized in Fig. 4, in which the amplitude and symmetry of the charge signal can be determined by the Néel vector **n**. Figure 4a, b show the schematics of ferroelastic switching of uniaxial magnetic anisotropy in PMN-PT/Mn$_2$Au. When a positive electric field $E_1$ ($+4\,\mathrm{kV\,cm^{-1}}$), which is larger than the ferroelectric coercive field on PMN-PT (011), is applied (Fig. 4a), n would be switched from [100] to [0$\bar{1}$1] (parallel to the $y$-axis)[13]. In contrast, when a negative $E_2$ ($-2\,\mathrm{kV\,cm^{-1}}$) with the opposite polarity up to the ferroelectric coercive field of PMN-PT is applied (Fig. 4b), the strain state and **n** would be aligned back to [100] (parallel to the $x$-axis). Note that such a switching is non-volatile, and all of the THz waveforms were measured after removing $E$.

The amplitude of THz waveforms from the Mn$_2$Au films as a function of the sample azimuth $\theta$ for the original state, $E_1$ state and $E_2$ state are summarized in Figs. 4c, 4d, and 4e, respectively (where $\theta$ is the sample in-plane rotation angle. When the Néel vector **n** is along

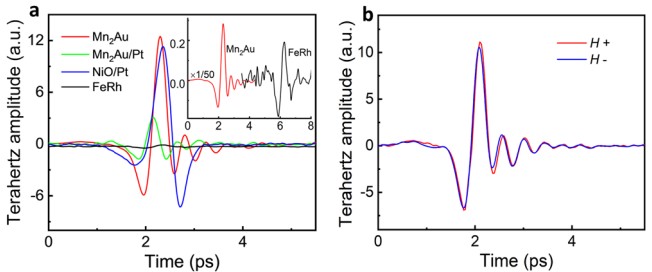

**Fig. 3 | THz emission for antiferromagnet (AFM) vs. AFM/heavy metal (HM) layers. a** A comparison of THz emission between Mn$_2$Au (red), Mn$_2$Au/Pt (purple), NiO/Pt (cyan), and FeRh (green). The pump fluence of the laser pulse is 12 uJ mm$^{-2}$ and the inset shows the very weak emission of FeRh and THz signal of Mn$_2$Au is scaled down for 50 times. **b** The THz emission from the Mn$_2$Au single layer shows no dependence on the magnetic field **H** (4 kOe).

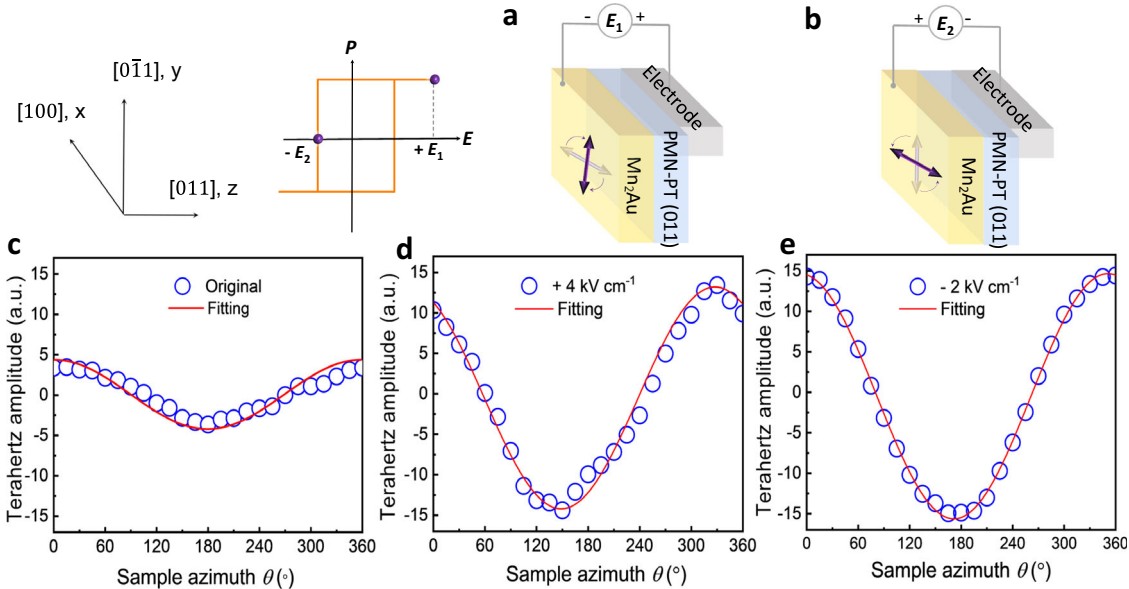

**Fig. 4 | Symmetric and amplitude of angular dependence of charge current generation. a, b** Schematic of ferroelastic strain switching of uniaxial magnetic anisotropy (UMA) driven by electric fields in Mn$_2$Au/PMN-PT (011) structure. Néel moments are switched toward the compressive direction of the PMN-PT substrate, [100] and [0$\bar{1}$1] axes in a) and b), respectively. The inset of **a** shows a representation of the $P$-$E$ loop, where $P$, $E_1$ ($+4\,\mathrm{kV\,cm^{-1}}$), and $E_2$ ($-2\,\mathrm{kV\,cm^{-1}}$) represent the electric polarization, positive electric field (larger than the ferroelectric coercive field), and negative electric field (coercive field), respectively. **c–e** The amplitude of THz waveforms as a function of sample azimuth $\theta$ with the original state (**c**), $E_1$ (**d**), $E_2$ (**e**), respectively.

[100] (Figs. 4c and 4e), the angular dependence of the amplitude is clearly consistent with the uniaxial magnetic anisotropy of $Mn_2Au$ Néel vector $\boldsymbol{n}$ with a period of 180° ($cos\theta$) with the maximum of the THz at $\theta = 0°$ and 360° and the minimum at $\theta = 180°$. The amplitude of the THz waveform after applying electric field is almost 3 times larger than that of the original state[34], revealing that the electric field clearly modulates the antiferromagnetic domain distribution of $Mn_2Au$, guaranteeing the Néel-type domains along the [100] direction more ordered compared to the original state, and the amplitude of the emitted THz signal is enhanced. In contrast, a different symmetry is found after applying a strain electric field $E_1$ (Fig. 4d). The Néel vector is rotated from [100] to [0$\bar{1}$1] (90° rotation). The most eminent feature is that the THz waveform data can be fitted by the cosine function with a 30° phase shift. The maximum THz amplitude appears at $\theta = 330°$, and the minimum appears at $\theta = 150°$. This phenomenon arises because the size of the laser spot used in our work is ~3 mm, which is much larger than the size of the Néel-type domains of $Mn_2Au$[35,36]. Especially, XMLD-PEEM directly exhibits that the domain switching is almost 30% with NSOT switching[36]. The results indicate that the proportion of the Néel vector reorientation in this sample is approximately 30% in the macroscopic region of the sample.

Similarly, the magnitude of the charge current in the original state is expected to be sample-dependent (See Supporting Information Note4), its symmetry is determined by both the magnetocrystalline anisotropy[37] and Néel vector of the sample. The resulting THz signal is emitted by the $y$-component of total charge current generation, and the charge current is perpendicular to the Néel vector $\boldsymbol{n}$, which verifies that the mechanism of charge current on satisfies the reciprocal relationship of NSOT.

In conclusion, we demonstrate the AFM magnonic charge current generation in collinear AFM metallic $Mn_2Au$ at room temperature via the reciprocal relationship of NSOT. In the THz emission experiments, the laser pump pulse triggers the AFM magnon fluctuation in thin films. Instantaneous spin polarization in the staggered magnetic sublattice is achieved, which pumps AFM magnonic charge that is converted into charge current directly. In addition to $Mn_2Au$, the AFM magnonic charge current generation is expected to exist in collinear AFM metallic systems with broken space-reversal symmetry, such as CuMnAs and $RuO_2$[38]. The generation of AFM magnonic charge current generation opens up new prospects for THz antiferromagnetic magnonics and sets the stage for future exploring of magnonic pumping in the emerging field of AFM spintronics.

## Methods

### Sample fabrication and characterization
15-nm-thick (103)-oriented $Mn_2Au$ films on ferroelastic crystals Pb(Mg$_{1/3}$Nb$_{2/3}$)$_{0.7}$Ti$_{0.3}$O$_3$ (PMN-PT) (011) substrates were deposited by magnetron sputtering at 573 K. The base pressure is $2 \times 10^{-5}$ Pa, and the growth rate is 0.07 nm/s using a $Mn_2Au$ alloy target (atomic ratio of 2:1). X-ray diffraction of the $Mn_2Au$ film was measured using Cu $K\alpha_1$ radiation with $\lambda = 1.5406$ Å, indicating a texture films with a (103) phase (See Supplementary Note5). The surface roughness was characterized by an atomic force microscope (AFM). Magnetic properties were measured by a superconducting quantum interference device (SQUID) magnetometry with a field of up to 5 Tesla (see Supplementary Note 6).

### THz emission spectroscopy
A commercial Ti:sapphire laser (central wavelength of 800 nm, pulse duration of 100 fs, repetition rate of 1 kHz) was used for THz emission measurements. The pumping laser beam was split into two parts (at a 9:1 ratio of intensities) for photogenerating and electro-optic sampling of the THz spin currents. The pumping laser pulses were focused onto the emission samples with a pot diameter of around 3 mm. The laser fluence was 12 μJ mm$^{-2}$ for most of the measurements. Our measurements were obtained with linearly polarized laser pulses, and the

emitted THz wave was collected and refocused by two parabolic mirrors with a reflected focal length of 5 cm. The THz electric field was temporally probed by measuring the ellipticity modulation of the probe beam in a 1-mm-thick (110)-oriented ZnTe crystal. All of the measurements were conducted at room temperature with dry air.

## Data availability
The dataset of the main figures generated in this study is provided in the Supplementary Information/Source Data file. Source data are provided with this paper.

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

## Acknowledgements

L. Huang., L. Liao, and H. Qiu contributed equally to this work. This work was supported by the National Natural Science Foundation of China (Grant Nos. 52225106 (C.S.), 52301243 (L.H.), 62171216 (H.Q.), and 12241404 (C.S.)), and the National Key Research and Development Program of China (Grant No. 2022YFA1402603 (C.S.)).

## Author contributions

C.S. and L.H. designed the experiment. L.H., L.L., and H.Q. performed the experiments and analyzed the data. L.H. provided the comparison samples. X.C., Y.Z., H.B., and Z.Z. contributed to discussion and theoretical interpretation of the results. The manuscript was written by L.H., L.L., and H.Q.; C.S., B.J., and F.P. supervised this study, with feedback and input from all the coauthors.

## Competing interests

The authors declare no competing interests.
