## [Peer Review File · Nature Communications]

Reviewers' Comments:

Reviewer #1:

Remarks to the Author:

Antiferromagnetic-based spintronics devices immune to external magnetic perturbation have immense applications for high-speed electronics. Transient terahertz frequency generation from ferromagnetic materials has recently drawn the attention of researchers and realizing the same through robust antiferromagnetic films is highly desirable. On a similar line, an article recently published, has demonstrated terahertz emission from NiO films [Hongsong et al. Nature Physics, 2020], and here the authors have investigated a different antiferromagnetic material Mn₂Au for its characteristic broken inversion symmetry.

I appreciate the author for a thorough study of Mn₂Au exhibiting potential applications in generating terahertz frequencies, using non-equilibrium photoexcitation, which otherwise demonstrates precession at 121 GHz under equilibrium excitation. Interestingly, a few co-authors in this work have earlier established the electric field control of Neel spin-orbit torque in Mn₂Au and that provides the basis for electrically controlled terahertz spin current [Chen et al. Nature Materials, 18, 931(2019)]. The main observations of the manuscript are:

1. Generating THz frequencies from ultrathin antiferromagnet, Mn₂Au by producing polarized spin current through photoexcitation and employing intrinsic broken-inversion symmetry to provide the spin-to-charge conversion.

2. The authors have observed electrical switching of the antiferromagnetic Neel vector and discussed the relationship between the polarization direction of emitted THz signal and the antiferromagnetic Neel vector.

The investigation, results, and explanation of the manuscript is concise, but I found the manuscript hard to follow. In my opinion, the manuscript warrants publication but requires a major revision. I have listed a few below but would like the authors to consider rewriting several parts. Moreover, there are certain areas that the authors need to clarify further, and I have a few comments concerning the paper, as explained below.

1. The authors have not adequately detailed the generation process of polarized spin current in AFM. Here, ideally, the spin current generation occurs through the laser-induced impulsive change in magnetic moment M .

Throughout the manuscript, the authors have highlighted the broken inversion symmetry of Mn₂Au, but that only explains the process of intrinsic spin-to-charge conversion. Can the authors include some discussion regarding the same and adequately explain the generation and spin-to-charge conversion process distinctly?

2. In my opinion, the Mn₂Au exhibits a layered structure of opposite spin orientation, and hence the overall magnetic moment compensates in even sets of layers. Can the polarized spin current generation be associated with the uncompensated magnetic moment due to odd sets of layers? A possible confirmation is in supplementary Figure S3, where the authors have periodically shown high and low THz emissions with increasing thickness.

Did the authors ensure layer-wise growth of Mn₂Au layers to provide insights into the possible explanation? Another insight can be drawn from the M-H curve, as shown in Figure S2. Can the authors please provide a zoom-in result of the M-H curve and discuss them in supplementary?

3. Throughout the manuscript, the authors have highlighted "charge pumping in the antiferromagnet," which rather should be written as "charge current generation in the antiferromagnet". Similarly, this also applies to the title of the manuscript.

4. At several places in the manuscript, the authors have provided weak statements such as,

"Antiferromagnetic (AFM) materials with the intrinsic frequency of terahertz (THz) and high stability against disturbing external fields are prime candidates to replace ferromagnets"

In my opinion, AFM can never replace FM in all its applications. However, AFM may replace FM for

a few applications.

“However, whether the charge current can be directly generated in AFM thin film is still unclear.”

Please refer : Chen, X., Zhou, X., Cheng, R. et al. Electric field control of Néel spin-orbit torque in an antiferromagnet. *Nat. Mater.* 18, 931–935 (2019)

“We demonstrate that the broadband feature of the spectrum is completely different from the narrowband (coherent) magnons (~ 121 GHz) at low frequency, suggesting that the mechanism of THz wave generated by the magnetic dipole radiation has been excluded.”

The magnetic dipole radiation cannot be excluded for THz emission. Refer : Zhang, W., Maldonado, P., Jin, Z. et al. Ultrafast terahertz magnetometry. *Nat Commun* 11, 4247 (2020).
<https://doi.org/10.1038/s41467-020-17935-6>

5. It is surprising to notice such a low THz pulse generation from Mn₂Au/Pt compared to Mn₂Au. Can the authors comment on the pronounced difference in the amplitude of these two cases?

The authors have not mentioned the thickness of Platinum in the manuscript; however, the data suggests a much higher thickness of platinum, maybe >10nm. It is recommended to use 2-4 nm of Platinum for the most optimized output. The authors have made a strong statement by comparing the THz output from Mn₂Au/Pt and Mn₂Au. Can the authors provide insight into the deposition of Platinum (and also include it in the manuscript)? Did they deposit Mn₂Au and Pt without breaking the vacuum during the whole process?

6. Why did they take 15nm of Mn₂Au for investigation, whereas they found the highest THz generation from 11nm Mn₂Au? Can the authors comment on the thickness-dependent THz pulse amplitude from Mn₂Au?

7. The authors have shown Azimuth dependent THz amplitude. It is not precisely clear to me how can rotating the sample cause change in terahertz amplitude. Does azimuth-dependent THz amplitude manifest due to the rotation of the sample with respect to the incident laser polarization?

Can the author also comment if the polarization of output terahertz is also changed, and how exactly did the author perform the measurements? If yes, note that the sensitivity of detector ZnTe crystal is different for different polarization of incoming THz wave. Did the author consider that? Can the author please provide the setup details in the figure of how the measurements in figure 4(c-e) were measured?

The authors should consider referencing a recent field control THz spintronic emission work *Nat. Commun.* 13, 4072 (2022).

Reviewer #2:

Remarks to the Author:

In this article, Lin Huang et al. present optical pump-THz emission measurements on Mn₂Au thin films grown on ferroelectric PMN-PT. Upon pumping by a linearly polarised optical pulse they observe broad-band THz emission and claim that its origin resides in magnonic charge pumping.

The evidence that the authors offer to justify this claim is however far too small.

- The authors keep referring to magnon precession and this is not justified. Optical pumping results in a highly non-equilibrium state in a metal. The process to reach equilibrium is highly dissipative, involving different scattering processes, so I don't see the reason why magnonic charge pumping should be justified at all. Instead the authors fail to discuss other effects that could arise because of symmetry breaking such as optical rectification, photogalvanic effect....

- In any case, it would be important to see how the emission depends on the polarisation of the pump, which is not shown. How are magnons generated?

- In Fig. 4 the authors show how the amplitude of the THz emission polarised along the y axis changes as the sample is rotated in-plane, observing a 360 degrees period. If the THz emission was connected to the Neel vector I would expect to see a 180 degrees period and a 90 degrees phase shift between the graphs in Fig.4d and Fig.4e, which the authors don't observe. They claim that the 30 degrees shift that they instead observe is caused by the presence of several domains and that "the switching of Neel-type domains is approximately 30% in the field of laser spot of 30 mm". But this should cause a reduction of the emission not a phase shift. Plus, why are the authors talking about switching?

- The thickness dependence presented in the SI is very curious and the authors should justify it much more extensively.

Reviewer #3:

Remarks to the Author:

Huang et al. report the observation of laser pulse generated THz-radiation emitted from Mn₂Au thin films. They assume that this effect is due to the generation of AFM magnon oscillations, which is assumed to result via an "reciprocal" Néel spin-orbit torque in an oscillating charge current.

However, there are several open questions which require clarifications before the validity of these assumptions can be assessed:

1) Please provide more X-ray diffraction (XRD) based evidence that the investigated sample is really Mn₂Au.

The weak XRD peak shown in Fig. S1 and labeled Mn₂Au(103) appears to be at a 2Theta angle of 40deg, whereas it should appear at 41.7deg based on Pearson's handbook of intermetallic compounds. Can you exclude that this peak is due to a pseudocubic MnAu phase with $a = 3.183\text{\AA}$ and $c = 3.283\text{\AA}$, which is mentioned in Pearson's handbook and is, e.g., discussed in Appl. Phys. Lett. 79, 1507 (2001)? The XRD peak position of 40deg would fit well to the (101) and (011)-peaks of this MnAu phase. It is absolutely required that XRD-data of out-of-plane peaks such as (002) and (200) of Mn₂Au is shown to make sure that really the intended compound is investigated. Are the sample ordered in-plane?

Furthermore, please provide a zoomed in representation of the XRD peak, which you label as Mn₂Au(103). What is the width of this peak in 2Theta? Does that correspond to the film thickness based on the Scherrer formula? Please provide also a rocking curve (omega-scan with fixed 2Theta) of this peak. What is its width, i.e., what is the degree of mosaicity of the samples? If no convincing evidence for an epitaxial Mn₂Au phase can be provided, all other issues discussed below are not relevant anymore and the paper should be rejected.

2) Regarding sample characterization, the results of other groups based on completely different Mn₂Au samples cannot be used to draw conclusions regarding the samples discussed here. E.g., the discussion of the antiferromagnetic domain size on page 8 cites work from another group based on epitaxial Mn₂Au(001) samples grown on different substrates. There is no reason to expect similar domains in the completely different samples discussed here. This applies also to the cited work on imaging of current driven domain wall switching on page 8. Please make a clear statement, that no microscopic magnetic characterization of the samples discussed here is available (or is it?).

3) It is beyond doubt that the Femtosecond laser pulse generates THz radiation from the sample. However, the role of the piezoelectric substrate in this process requires further investigations. The laser pulse heats the substrate, which is resulting in anisotropic thermal expansion. This corresponds to the formation of a piezoelectric voltage, which should drive a current in the metal film on top. This could be the origin of THz emission independent from the specific metallic material on top, i.e., could be not related to antiferromagnetism at all. Please provide reference experiments with a corresponding non-magnetic metal thin film grown on

your PNM-PT substrates.

4) The abstract and introduction suggests, that the existence of a significant Néel spin-orbit torque in Mn₂Au is experimentally established, which is not the case. It should be mentioned that there is strong evidence for dominant thermomagnetoelastic switching in other antiferromagnets (e.g. Nano Lett. 2021, 21, 1, 114–119). Regarding current driven Néel vector switching in Mn₂Au, it is still an open question if NSOT or thermomagnetoelastic mechanisms are stronger. However, regarding potential laser pulse induced heating effects associated with the experiments discussed by the authors, a possible thermomagnetoelastic effect on the Néel vector is highly relevant and should be considered.

5) The authors explain the laser pulse induced THz emission by a “reciprocal” NSOT. Please provide more information about the concept of this mechanism. The introduction describes an SOT generated at the interface between a heavy metal and an antiferromagnet, which is only loosely related to bulk NSOT and therefore misleading. Please explain the NSOT mechanism instead. Also in the introduction, it is mentioned that a spin current is generated by “antiferromagnetic dynamics”. What is meant here, a spin current carried by magnons? Please provide information on how the laser pulse couples to antiferromagnetic magnons in a way that a staggered magnetic field is generated at the antiferromagnetic sublattices of Mn₂Au, as is would be required for a (reciprocal) NSOT mechanism.

Response Letter of NCOMMS-23-00324

We have addressed the issues raised by three referees point by point below. The amendments in our revised manuscript are highlighted in bold face style. The main revisions include:

- 1) We provide detailed evidence and explanations of **identifying the material as Mn₂Au** in the first response of reviewer 3.
- 2) We added the description of **the Schematic of THz emission measurement setup in supporting information S1**.
- 3) We added the description of **the magnonic charge current generation process in supporting information S2**.
- 4) We revised the description of **the thickness dependent emission in supporting information S5**.
- 5) We added the description of **Effect of heavy metals on terahertz emission in supporting information S6**.
- 6) We provided detailed explanations of the **Néel vector switching** in the third response of **reviewer 2** and the fourth question of **reviewer 3**.
- 7) We revised the “charge pumping” to “charge current generation” in our manuscript. The title is changed as **“Antiferromagnetic magnonic charge current generation via ultrafast optical excitation”**.
- 8) We revised the explanation of the **NSOT mechanism** in the **Introduction** part.
- 9) The controlled THz emission experiment performed with in a corresponding 10 nm-thick non-magnetic metal thin film Pt film (10 nm) grown on PMN-PT (011) substrates is provided in **Fig. R7**.

Response to Reviewer #1:

Antiferromagnetic-based spintronics devices immune to external magnetic perturbation have immense applications for high-speed electronics. Transient terahertz frequency generation from ferromagnetic materials has recently drawn the attention of researchers and realizing the same through robust antiferromagnetic films is highly desirable. On a similar line, an article recently published, has demonstrated terahertz emission from NiO films [Hongson et al. Nature Physics, 2020], and here the authors have investigated a different antiferromagnetic material Mn₂Au for its characteristic broken inversion symmetry.

I appreciate the author for a thorough study of Mn₂Au exhibiting potential applications in generating terahertz frequencies, using non-equilibrium

photoexcitation, which otherwise demonstrates precession at 121 GHz under equilibrium excitation. Interestingly, a few co-authors in this work have earlier established the electric field control of Neel spin-orbit torque in Mn₂Au and that provides the basis for electrically controlled terahertz spin current [Chen et al. Nature Materials, 18, 931(2019)]. The main observations of the manuscript are:

1. Generating THz frequencies from ultrathin antiferromagnet, Mn₂Au by producing polarized spin current through photoexcitation and employing intrinsic broken-inversion symmetry to provide the spin-to-charge conversion.
2. The authors have observed electrical switching of the antiferromagnetic Neel vector and discussed the relationship between the polarization direction of emitted THz signal and the antiferromagnetic Neel vector.

The investigation, results, and explanation of the manuscript is concise, but I found the manuscript hard to follow. In my opinion, the manuscript warrants publication but requires a major revision. I have listed a few below but would like the authors to consider rewriting several parts. Moreover, there are certain areas that the authors need to clarify further, and I have a few comments concerning the paper, as explained below.

Reply: We are grateful to the reviewer for carefully reviewing our manuscript and positive evaluation “I appreciate the author for a thorough study of Mn₂Au exhibiting potential applications in generating terahertz frequencies, using non-equilibrium photoexcitation.....”

1. The authors have not adequately detailed the generation process of polarized spin current in AFM. Here, ideally, the spin current generation occurs through the laser-induced impulsive change in magnetic moment M .

Throughout the manuscript, the authors have highlighted the broken inversion symmetry of Mn₂Au, but that only explains the process of intrinsic spin-to-charge conversion. Can the authors include some discussion regarding the same and adequately explain the generation and spin-to-charge conversion process distinctly?

Reply: The oscillation of the Néel vector n in Mn₂Au thin film is excited by the ultrafast optical pulse, resulting in locally non-equilibrium spin polarization with opposite signs on the antiferromagnetic sublattices. The spin-to-charge conversion can be described $J_c \sim \sigma_{A,B} \times \hat{z}$ by the reciprocal effect of NSOT and leads to THz radiation. A more detailed discussion of the process and

theoretical model please see Supporting Information S1. For clarity, we revised the introduction parts in the manuscript (Page 3 Line 13-16): **The femtosecond laser pulse triggers the Néel vector oscillation in thin films, resulting in locally non-equilibrium spin polarization ($\sigma_A = -\sigma_B$) with opposite signs on the Mn sublattices (See Supporting Information S2).**

We add the description of the charge current generation process in supporting information S2:

S2 Néel spin-orbit torques and magnonic charge current generation

It is well established that charge current-induced local spin polarization can couple to the local magnetic moment to produce Néel spin-orbit torque (NSOT), which enables a current-induced rotation of the local moments while preserving their Néel vector [1]. Accordingly, in the reciprocal process of NSOT, a charge current is expected to occur in AFM with local inversion symmetry breaking when staggered non-equilibrium spin polarization is excited by antiferromagnetic magnetization dynamic, i.e., a fluctuation of the Néel vector n .

To identify the THz emission mechanisms in Mn_2Au thin films, we measured the time domain THz signals for different linear pump polarization angle α , as shown in Fig. S2. The THz emission signal exhibits independence of the linear pump polarization angle α (**Fig. S2a**), where the azimuth θ of the sample remains the same (see Supporting Information S1 for THz emission measurement setup). Accordingly, **Fig. S2b** shows an isotropic behavior of the THz signal amplitude on the linear pump polarization (angle α), in contrast to a recent report in (111) NiO thin films [2]. This result suggests that thermal absorption in the metallic material Mn_2Au is the driving mechanism of ultrafast spin-current generation with laser pump pulse irradiating the sample [3].

Fig. S2 (a) Time domain THz emission from Mn_2Au with variation of the linear pump polarization angle α . (b) Dependence of the THz signal on the linear pump

polarization (angle α) showing an isotropic behavior. Error bars correspond to standard deviations.

In the crystalline structure of Mn_2Au (**Fig. S3a**), the magnetic sublattices exhibit a locally broken inversion symmetry, and the optical pump pulse irradiates the sample surface. The oscillation of the Néel vector \mathbf{n} is excited by the ultrafast optical pulse, resulting in locally non-equilibrium spin polarization with opposite signs on the sublattices. From the initial state illustrated in **Fig. S3b**, Néel vector \mathbf{n}_0 is in the initial state before the optical pump irradiation, which is represented in purple. When the optical pump is applied, a deflection of the Néel vector $\mathbf{n}(t)$ followed by the oscillation, i.e., magnonic excitation is generated (**Fig. S3c**), due to the thermo-magnetoelastic effect [4] or electron thermal excitation [5] of the metal layer Mn_2Au .

The Holstein-Primakoff transformation for antiferromagnetic magnons indicates spins on site i as $S_{A,i}^n = S_0 - a_i^+ a_i$, $S_{B,i}^n = -S_0 + b_i^+ b_i$. Here, n labels the equilibrium Néel vector direction, A and B label the sublattices, a_i, a_i^+, b_i, b_i^+ are the annihilation and creation operators for site i at A and B sublattices, respectively, S_0 is the spin of the moment [6]. When magnon is excited, the spins on A and B sublattices derive from $\pm S_0$, result in the local spin polarization ($\sigma_A = -\sigma_B$), which is equal to the deviations $\sigma\mathbf{n}$ of the Néel vector from the equilibrium \mathbf{n}_0 . Thus, a charge current can be described as $J_c \sim \sigma_{A,B} \times \hat{\mathbf{z}}$ by the reciprocal effect of NSOT, and leads to THz radiation.

Fig. S3. Generation of magnonic charge current. (a) The crystalline structure of Mn_2Au . Magnetic sublattices Mn_A and Mn_B are marked in red and green, respectively. The projection of magnetic sublattices Mn_A and Mn_B on the Mn_2Au (103) plane aligns along the $[\bar{3}31]$ direction, and laser irradiates on the sample surface. (b) The initial state of Néel vector \mathbf{n}_0 is represented in purple. (c) Via laser irradiation, a deflection of the Néel vector happens followed by oscillations, i.e. magnon excitations. A magnonic charge current is generated on the Mn_2Au surface by the reciprocal of NSOT, and leads to THz radiation.

2. In my opinion, the Mn_2Au exhibits a layered structure of opposite spin orientation,

and hence the overall magnetic moment compensates in even sets of layers. Can the polarized spin current generation be associated with the uncompensated magnetic moment due to odd sets of layers? A possible confirmation is in supplementary Figure S3, where the authors have periodically shown high and low THz emissions with increasing thickness.

Did the authors ensure layer-wise growth of Mn₂Au layers to provide insights into the possible explanation? Another insight can be drawn from the M-H curve, as shown in Figure S2. Can the authors please provide a zoom-in result of the M-H curve and discuss them in supplementary?

Reply: We believe that the intensity of the terahertz signal emitted by the sample is determined by a random arrangement and distribution of the antiferromagnetic domains of the film, regardless of the thickness. Therefore, we prepared samples with different thicknesses (11 nm and 15 nm), where four wafers for the growth of the 11 nm-thick samples were prepared simultaneously, also two samples for the 15 nm were prepared at the same time. Through terahertz emission experiments, it shows significant differences in the intensity and polarity of terahertz emission in the original state for the four 11 nm samples (**Fig. S6a**). Such a difference was also found in two 15 nm samples (**Fig. S6b**). However, after applying an electric field (+4 kV cm⁻¹) to the samples, the terahertz signal intensity of all the samples prepared simultaneously keeps almost the same (11 nm in **Fig. S7a** and 15 nm in **Fig. S7b**). It indicates that the generation of polarized spin is independent of the uncompensated magnetic moment due to the odd set of layers in the original state. We show the enlarged view of the M-H curve from PMN-PT/Mn₂Au films in **Fig. R1**. Only the diamagnetic background from the PMN-PT substrate can be detected, which indicates the antiferromagnetic characteristic of Mn₂Au.

Fig. R1. The enlarged view of the M-H curve from PMN-PT/Mn₂Au films.

We revised the Result parts in the manuscript (Page 8 Line 22-24): **Similarly, the magnitude of the charge current in the original state is expected to be sample-dependent (See Supporting Information S5), its symmetry is determined by both the magnetocrystalline anisotropy³³ and Néel vector of the sample.**

And we add the description of the charge current generation process in supporting information S5:

S5 Thickness dependence of THz emission

The intensity of the terahertz signal emitted by the sample is determined by a random arrangement and distribution of the antiferromagnetic domains of the film, regardless of the thickness. Therefore, we prepared samples with different thicknesses (11 nm and 15 nm), where four samples for the growth of the 11 nm-thick samples were prepared simultaneously, also two samples for the 15 nm were prepared at the same time. Through terahertz emission experiments, it shows significant differences in the intensity and polarity of terahertz emission in the original state for the four 11 nm samples (**Fig. S6a**). Such a difference was also found in two 15 nm samples (**Fig. S6b**). However, after applying an electric field ($+4 \text{ kV cm}^{-1}$) to the samples, the terahertz signal intensity of all the samples prepared simultaneously keeps almost the same (11 nm in **Fig. S7a** and 15 nm in **Fig. S7b**). It indicates that the generation of polarized spin is independent of the uncompensated magnetic moment due to the odd set of layers in the original state.

Fig. S6. (a) Terahertz emission for four 11 nm samples. (b) Terahertz emission for two 15 nm samples.

Fig. S7. After applying $+4 \text{ kV cm}^{-1}$ to the wafers, terahertz absolute signal of (a) four 11 nm samples and (b) two 15 nm samples.

3. Throughout the manuscript, the authors have highlighted “charge pumping in the antiferromagnet,” which rather should be written as “charge current generation in the antiferromagnet”. Similarly, this also applies to the title of the manuscript.

Reply: We replace “charge pumping” with “**charge current generation**” in our manuscript. The title is changed as “**Antiferromagnetic magnonic charge current generation via ultrafast optical excitation**”.

4. At several places in the manuscript, the authors have provided weak statements such as,

“Antiferromagnetic (AFM) materials with the intrinsic frequency of terahertz (THz) and high stability against disturbing external fields are prime candidates to replace ferromagnets”

In my opinion, AFM can never replace FM in all its applications. However, AFM may replace FM for a few applications.

“However, whether the charge current can be directly generated in AFM thin film is still unclear.”

Please refer : Chen, X., Zhou, X., Cheng, R. et al. Electric field control of Néel spin–orbit torque in an antiferromagnet. *Nat. Mater.* 18, 931–935 (2019) 13

“We demonstrate that the broadband feature of the spectrum is completely different from the narrowband (coherent) magnons ($\sim 121 \text{ GHz}$) at low frequency, suggesting that the mechanism of THz wave generated by the magnetic dipole radiation has been excluded.”

The magnetic dipole radiation cannot be excluded for THz emission. Refer : Zhang, W., Maldonado, P., Jin, Z. et al. Ultrafast terahertz magnetometry. Nat Commun 11, 4247 (2020). <https://doi.org/10.1038/s41467-020-17935-6>

Reply: We revised the description of the Introduction part in Page 2 Line 2-4: **Antiferromagnets (AFM) with the intrinsic frequency of terahertz (THz) and high stability against disturbing external fields, are prime candidates to develop new types of memory and logic devices¹⁻³**. And we cite the Ref. Nat. Mater. 18, 931–935 (2019) as Ref. 13 in the sentence “**However, whether the charge current can be directly generated in AFM single film is still unclear¹³.**”

The paper Nat Commun 11, 4247 (2020) studied ultrafast magnetization dynamics in a laser-excited ferromagnetic iron films, which the measured electric field component of the laser-driven magnetic dipole THz emission from the sample. Terahertz radiation caused by magnetic dipoles in such ferromagnetic metal materials cannot be ignored. In antiferromagnetic materials, there is also a very weak THz signal from magnetic dipoles, such as NiO and FeRh. Therefore, in order to avoid misunderstandings caused by our description, we cite this paper as **Ref.28** and revised our manuscript in Page 6 Line 16-18: **It suggests that the mechanism of the magnetic dipole radiation²⁸ as THz wave emission from Mn₂Au thin films can be excluded.**

5. It is surprising to notice such a low THz pulse generation from Mn₂Au/Pt compared to Mn₂Au. Can the authors comment on the pronounced difference in the amplitude of these two cases?

The authors have not mentioned the thickness of Platinum in the manuscript; however, the data suggests a much higher thickness of platinum, maybe >10nm. It is recommended to use 2-4 nm of Platinum for the most optimized output. The authors have made a strong statement by comparing the THz output from Mn₂Au/Pt and Mn₂Au. Can the authors provide insight into the deposition of Platinum (and also include it in the manuscript)? Did they deposit Mn₂Au and Pt without breaking the vacuum during the whole process?

Reply: The Mn₂Au layers in different samples of Mn₂Au (15 nm) and Mn₂Au (15 nm)/Pt (5 nm) were grown simultaneously to ensure that the domain distribution of the two control samples is consistent, so the vacuum has to be broken for

fetching a part of Mn₂Au single layer samples before the 5 nm-thick Pt deposition on the other part. We prepared a control sample for 10 nm-thick Mn₂Au, Mn₂Au (10 nm)/Pt (5 nm) and Mn₂Au (10 nm)/W (5 nm). The heavy metals Pt and W have opposite spin Hall angles. However, the polarity of terahertz emission does not reverse but decrease (**Fig. S8**). Therefore, we believe that Mn₂Au/heavy metal bilayer does not generate spins flowing into heavy metals, and the emission of terahertz signals due to the inverse spin Hall effect at the interface does not contribute. After covering the heavy metal layer, the terahertz signal from the bi-layer decreases because of the absorption of the heavy metal layer.

We add the description of the THz absorption of platinum in the manuscript in Page 6 Line 5–8: **It shows that the amplitude of THz signal from the Mn₂Au single layer is much higher than that of the Mn₂Au/Pt bilayer because the THz absorption of heavy metal layer decreases the terahertz signal (See Supporting Information S6).**

We add the detailed description in supporting information S6:

S6 Effect of heavy metals on terahertz emission

We prepared a control sample for 10 nm-thick Mn₂Au, Mn₂Au (10 nm)/Pt (5 nm) and Mn₂Au (10 nm)/W (5 nm). The heavy metals Pt and W have opposite spin Hall angles. However, the polarity of terahertz emission does not reverse but decrease (**Fig. S8**). Therefore, we believe that Mn₂Au/heavy metal bilayer does not generate spins flowing into heavy metals, and the emission of terahertz signals due to the inverse spin Hall effect at the interface does not contribute. After covering the heavy metal layer, the terahertz signal from the bi-layer decreases because of the absorption of the heavy metal layer.

Fig. S8 Terahertz time-domain spectrum of Mn₂Au (10 nm), Mn₂Au (10 nm)/Pt (5 nm) and Mn₂Au (10 nm)/W (5 nm).

6. Why did they take 15nm of Mn₂Au for investigation, whereas they found the highest THz generation from 11nm Mn₂Au? Can the authors comment on the thickness-dependent THz pulse amplitude from Mn₂Au?

Reply: In the original state, the terahertz wave signal radiated by the 15 nm sample is indeed weaker than its 11 nm counterpart. However, when the electric field is applied to the two samples, the effect of the electric field on the magnetic domain reorientation is much more obvious for the 15 nm film. This is most likely ascribed to higher crystalline quality for comparatively thicker films. For the 15 nm sample (**Fig. R2a**), the amplitude of the THz waveform after applying electric field (+4 kV cm⁻¹) is almost 3 times larger than that of the original state, revealing that the electric field clearly modulates the antiferromagnetic domain distribution of Mn₂Au, guaranteeing the Néel-type domains along the [100] direction more ordered compared to the original state, and the amplitude of the emitted THz signal is enhanced. Differently, the modulation with the electric field is much weaker for the 11 nm sample (**Fig. R2b**). The thickness-dependent THz pulse amplitude from Mn₂Au is described in detail in our answer to **Question 2**.

Fig. R2 Amplitude of THz waveforms after electric field applying on 15 nm (a) and 11 nm thin films (b).

7. The authors have shown Azimuth dependent THz amplitude. It is not precisely clear to me how can rotating the sample cause change in terahertz amplitude. Does azimuth-dependent THz amplitude manifest due to the rotation of the sample with respect to the incident laser polarization?

Can the author also comment if the polarization of output terahertz is also changed, and how exactly did the author perform the measurements? If yes, note that the sensitivity of detector ZnTe crystal is different for different polarization of incoming THz wave. Did the author consider that? Can the author please provide the setup details in the figure of how the measurements in figure 4(c-e) were measured?

The authors should consider referencing a recent field control THz spintronic emission work Nat. Commun. 13, 4072 (2022).

Reply: The schematic of the terahertz emission setup is shown in Supporting Information **Fig. S1**. The polarization of the pump laser is set along the y -axis. The polarization of the laser-induced THz wave rotates with the rotation of the sample, which is indicated by the angle γ . The y -component of the THz electric field is measured via electro-optical sampling by combining a wire grid polarizer and a (110)-cut ZnTe crystal.

The paper (Nat. Commun. 13, 4072 (2022)) studied electric-field control of terahertz spin current-based emitters on ferromagnetic/heavy metal bilayers. The nonlinear electric-field control of the spins occurs due to the strain-induced change in magnetic energy of the ferromagnet thin-film. This phenomenon is

somehow involved in our work. We cite this article in **Ref.30** and revised our manuscript in Page 8 Line 8-9: **The amplitude of the THz waveform after applying electric field is almost 3 times larger than that of the original state**³⁰...

And we add the description of the charge current generation process in supporting information S1:

S1 Schematic of THz emission measurement setup

The schematic of the terahertz emission setup is shown in **Fig. S1**, where WGP is wire-grid polarizer. The linear pump laser polarization angle α is defined with respect to y for the thin film. The pump laser propagates along the z -axis and sample is set in the x - y plane and the azimuth is denoted by θ . The polarization of the laser-induced THz wave rotates with the rotation of the sample, which is indicated by the angle γ . The y -component of the THz electric field is measured via electro-optical sampling by combining a wire grid polarizer and a (110)-cut ZnTe crystal.

Fig. S1 Schematic of terahertz emission setup, WGP: wire-grid polarizer.

Response to Reviewer #2:

In this article, Lin Huang et al. present optical pump-THz emission measurements on Mn₂Au thin films grown on ferroelectric PMN-PT. Upon pumping by a linearly polarised optical pulse they observe broad-band THz emission and claim that its origin resides in magnonic charge pumping.

The evidence that the authors offer to justify this claim is however far too small.

1- The authors keep referring to magnon precession and this is not justified. Optical pumping results in a highly non-equilibrium state in a metal. The process to reach equilibrium is highly dissipative, involving different scattering processes, so I don't see the reason why magnonic charge pumping should be justified at all. Instead the authors fail to discuss other effects that could arise because of symmetry breaking such as optical rectification, photogalvanic effect....

Reply: We are grateful to the reviewer for carefully reviewing our manuscript. We believe that the terahertz emitted from Mn₂Au is caused by the spin (magnon) and the distribution of the Néel-type domains of Mn₂Au determines the intensity of terahertz signal, regardless of the optical rectification and photogalvanic effect. Therefore, we prepared samples with different thickness (11 nm and 15 nm), where four 11 nm samples were prepared simultaneously, while two 15 nm samples were prepared at the same time. Through terahertz emission experiments, it shows significant differences in the intensity and polarity of terahertz emission in the original state for four 11 nm-thick samples (**Fig. S6a**). Similarly, this difference was found in two 15 nm samples (**Fig. S6b**). However, after applying an electric field (+4 kV cm⁻¹) to the samples, the terahertz signal intensity of the samples changes to almost the same (**Fig. S7**).

If the terahertz radiation is induced by symmetry breaking, such as optical rectification or photogalvanic effect, the terahertz signal emitted from samples of the same thickness should be consistent. Thus, we believe that the terahertz emission coming from Mn₂Au is induced by the spin (magnon) rather than the optical rectification or photogalvanic effect. On the other hand, the uncertainty of the amplitude of the THz signal between different samples is highly consistent with the random distribution of the spin-domain orientations along the easy axes in the Mn₂Au film. The domain of the thin film does not appear as a single domain in the laser range of 3 mm. The measured THz signals are

contributed by all the spin domains within the laser spot. Therefore, the inevitable uncertainty of the spin-domain orientation during sample preparation leads to the random change of the THz signal.

We revised the Result parts in the manuscript (Page 8 Line 22-24): **Similarly, the magnitude of the charge current in the original state is expected to be sample-dependent (See Supporting Information S5), its symmetry is determined by both the magnetocrystalline anisotropy³³ and Néel vector of the sample.**

And we add the description of the charge current generation process in supporting information S5:

S5 Thickness dependence of THz emission

The intensity of the terahertz signal emitted by the sample is determined by a random arrangement and distribution of the antiferromagnetic domains of the film, regardless of the thickness. Therefore, we prepared samples with different thicknesses (11 nm and 15 nm), where four samples for the growth of the 11 nm-thick samples were prepared simultaneously, also two wafers for the 15 nm were prepared at the same time. Through terahertz emission experiments, it shows significant differences in the intensity and polarity of terahertz emission in the original state for the four 11 nm samples (**Fig. S6a**). Such a difference was also found in two 15 nm samples (**Fig. S6b**). However, after applying an electric field ($+4 \text{ kV cm}^{-1}$) to the samples, the terahertz signal intensity of all the samples prepared simultaneously keeps almost the same (11 nm in **Fig. S7a** and 15 nm in **Fig. S7b**). It indicates that the generation of polarized spin is independent of the uncompensated magnetic moment due to the odd set of layers in the original state.

Fig. S6. (a) Terahertz emission for four 11 nm samples. (b) Terahertz emission for two 15 nm samples.

Fig. S7. After applying $+4 \text{ kV cm}^{-1}$ to the wafers, terahertz absolute signal of (a) four 11 nm samples and (b) two 15 nm samples.

2- In any case, it would be important to see how the emission depends on the polarisation of the pump, which is not shown. How are magnons generated?

Reply: We revised the description of the introduction parts in the manuscript (Page 3 Line 13-16): **The femtosecond laser pulse triggers the Néel vector oscillation in thin films, resulting in locally non-equilibrium spin polarization ($\sigma_A = -\sigma_B$) with opposite signs on the Mn sublattices (See Supporting Information S2).**

And we add the description of the charge current generation process in supporting information S2:

S2 Néel spin-orbit torques and magnonic charge current generation

It is well established that charge current-induced local spin polarization can couple to the local magnetic moment to produce Néel spin-orbit torque (NSOT), which enables a current-induced rotation of the local moments while preserving

their Néel vector [1]. Accordingly, in the reciprocal process of NSOT, a charge current is expected to occur in AFM with local inversion symmetry breaking when staggered non-equilibrium spin polarization is excited by antiferromagnetic magnetization dynamic, i.e., a fluctuation of the Néel vector \mathbf{n} .

To identify the THz emission mechanisms in Mn₂Au thin films, we measured the time domain THz signals for different linear pump polarization angle α , as shown in Fig. S2. The THz emission signal exhibits independence of the linear pump polarization angle α (**Fig. S2a**), where the azimuth θ of the sample remains the same (see Supporting Information S1 for THz emission measurement setup). Accordingly, **Fig. S2b** shows an isotropic behavior of the THz signal amplitude on the linear pump polarization (angle α), in contrast to a recent report in (111) NiO thin films [2]. This result suggests that thermal absorption in the metallic material Mn₂Au is the driving mechanism of ultrafast spin-current generation with laser pump pulse irradiating the sample [3].

Fig. S2 (a) Time domain THz emission from Mn₂Au with variation of the linear pump polarization angle α . (b) Dependence of the THz signal on the linear pump polarization (angle α) showing an isotropic behavior. Error bars correspond to standard deviations.

In the crystalline structure of Mn₂Au (**Fig. S3a**), the magnetic sublattices exhibit a locally broken inversion symmetry, and the optical pump pulse irradiates the sample surface. The oscillation of the Néel vector \mathbf{n} is excited by the ultrafast optical pulse, resulting in locally non-equilibrium spin polarization with opposite signs on the sublattices. From the initial state illustrated in **Fig. S3b**, Néel vector \mathbf{n}_0 is in the initial state before the optical pump irradiation, which is represented in purple. When the optical pump is applied, a deflection of the Néel vector $\mathbf{n}(t)$ followed by the oscillation, i.e., magnonic excitation is generated (**Fig. S3c**), due to the thermo-magnetoelastic effect [4] or electron thermal excitation [5] of the metal layer Mn₂Au.

The Holstein-Primakoff transformation for antiferromagnetic magnons

indicates spins on site i as $S_{A,i}^n = S_0 - a_i^+ a_i, S_{B,i}^n = -S_0 + b_i^+ b_i$. Here, n labels the equilibrium Néel vector direction, A and B label the sublattices, a_i, a_i^+, b_i, b_i^+ are the annihilation and creation operators for site i at A and B sublattices, respectively, S_0 is the spin of the moment [6]. When magnon is excited, the spins on A and B sublattices derive from $\pm S_0$, result in the local spin polarization ($\sigma_A = -\sigma_B$), which is equal to the deviations σn of the Néel vector from the equilibrium n_0 . Thus, a charge current can be described as $J_c \sim \sigma_{A,B} \times \hat{z}$ by the reciprocal effect of NSOT, and leads to THz radiation.

Fig. S3. Generation of magnonic charge current. (a) The crystalline structure of Mn_2Au . Magnetic sublattices Mn_A and Mn_B are marked in red and green, respectively. The projection of magnetic sublattices Mn_A and Mn_B on the Mn_2Au (103) plane aligns along the $[\bar{3}31]$ direction, and laser irradiates on the sample surface. (b) The initial state of Néel vector n_0 is represented in purple. (c) Via laser irradiation, a deflection of the Néel vector happens followed by oscillations, i.e. magnon excitations. A magnonic charge current is generated on the Mn_2Au surface by the reciprocal of NSOT, and leads to THz radiation.

3- In Fig. 4 the authors show how the amplitude of the THz emission polarised along the y axis changes as the sample is rotated in-plane, observing a 360 degrees period. If the THz emission was connected to the Neel vector I would expect to see a 180 degrees period and a 90 degrees phase shift between the graphs in Fig.4d and Fig.4e, which the authors don't observe. They claim that the 30 degrees shift that they instead observe is caused by the presence of several domains and that "the switching of Neel-type domains is approximately 30% in the field of laser spot of 30 mm". But this should cause a reduction of the emission not a phase shift. Plus, why are the authors talking about switching?

Reply: In general, the THz signal is magneto-optical signal, so magnetic symmetry should simultaneously obey the magnetocrystalline anisotropy (Phys. Rev. B 104, 024401 (2021)) and Néel vector symmetry of the sample. In our

manuscript, we emphasize the Néel vector dependence of THz emission, but the symmetry of the terahertz signal is related to the combination of the two. The (103)-phase Mn_2Au shows a uniaxial magnetocrystalline anisotropic (Supporting Information in Ref. Nat. Mater. 18, 931–935 (2019)). Although the symmetry of the Néel vector is 180° , the terahertz magneto-optical signal shows an overall period of 360° , taking both the symmetry of the magnetocrystalline anisotropic and Néel vector into account.

The size of the laser spot used in our work is ~ 3 mm, which is much larger than the size of the Néel-type domains of Mn_2Au (Phys. Rev. B 97, 134429 (2018) and Phys. Rev. B 99, 140409 (2019)). On the other hand, the antiferromagnetic domain switching is a multi-domain switching, as revealed by the transport and domain characterization (Science 351, 587–590 (2016), Nat. Mater. 18, 931–935 (2019), and Appl. Phys. Lett. 117, 082401 (2020)). Especially, XMLD-PEEM directly exhibits that the domain switching is almost 30% with NSOT switching (Phys. Rev. B 99, 140409 (2019)). Thus, when we modulate the antiferromagnetic Néel-type domains (Néel vector) by electric field, there is a 30% modulation of the 360° periodic curve. From **Fig.R3**, after applying different electric field to the same sample, the terahertz emission of the same sample azimuth varies by 30%. Therefore, the total change is 30%, which is reflected in the function of terahertz intensity with azimuth angle of $\cos(\theta+30)$. We mentioned in the manuscript that the electric field is applied to switch the Néel vector, then pump laser is used to irradiate the sample to achieve the oscillation of Néel vector, while radiating the terahertz signal.

Fig.R3. THz waveforms as sample azimuth at 0° after applying electric fields $+ 4 \text{ kV cm}^{-1}$ and $- 2 \text{ kV cm}^{-1}$.

We revised the manuscript (Page 8 Line 16–21): **This phenomenon arises because the size of the laser spot used in our work is ~3 mm, which is much larger than the size of the Néel-type domains of Mn₂Au^{31,32}. Especially, XMLD-PEEM directly exhibits that the domain switching is almost 30% with NSOT switching³². The results indicate that the proportion of the Néel vector reorientation in this sample is approximately 30% in the macroscopic region of the sample.**

We revised the Result part in the manuscript (Page 8 Line 22-24): **Similarly, the magnitude of the charge current in the original state is expected to be sample-dependent (See Supporting Information S5), its symmetry is determined by both the magnetocrystalline anisotropy³⁴ and Néel vector of the sample.**

4- The thickness dependence presented in the SI is very curious and the authors should justify it much more extensively.

Reply: The thickness-dependence of THz pulse amplitude from Mn₂Au is described in our answer to **Question 1**.

Response to Reviewer #3:

Huang et al. report the observation of laser pulse generated THz-radiation emitted from Mn₂Au thin films. They assume that this effect is due to the generation of AFM magnon oscillations, which is assumed to result via an “reciprocal” Néel spin-orbit torque in an oscillating charge current.

However, there are several open questions which require clarifications before the validity of these assumptions can be assessed:

1) Please provide more X-ray diffraction (XRD) based evidence that the investigated sample is really Mn₂Au. The weak XRD peak shown in Fig. S1 and labeled Mn₂Au(103) appears to be at a 2Theta angle of 40deg, whereas it should appear at 41.7deg based on Pearson’s handbook of intermetallic compounds. Can you exclude that this peak is due to a pseudocubic MnAu phase with $a = 3.183\text{Å}$ and $c = 3.283\text{Å}$, which is mentioned in Pearson’s handbook and is, e.g., discussed in Appl. Phys. Lett. 79, 1507 (2001)? The XRD peak position of 40deg would fit well to the (101) and (011)-peaks of this MnAu phase. It is absolutely required that XRD-data of out-of-plane peaks such as (002) and

(200) of Mn₂Au is shown to make sure that really the intended compound is investigated. Are the sample ordered in-plane?

Furthermore, please provide a zoomed in representation of the XRD peak, which you label as Mn₂Au (103). What is the width of this peak in 2Theta? Does that correspond to the film thickness based on the Scherrer formula? Please provide also a rocking curve (omega-scan with fixed 2Theta) of this peak. What is its width, i.e., what is the degree of mosaicity of the samples?

If no convincing evidence for an epitaxial Mn₂Au phase can be provided, all other issues discussed below are not relevant anymore and the paper should be rejected.

Reply: We are grateful to the reviewer for carefully reviewing our manuscript and we are very confident that our samples are Mn₂Au. We have re-measured XRD of Mn₂Au/PMN-PT samples and obtained a stronger Mn₂Au (103) peak (**Fig. R4a**). The FWHM of this peak is 1.2°, and the particle size calculated by the Scherrer formula is ~7 nm. Corresponding rocking curve (omega-scan with fixed 2Theta) is shown in **Fig. R4b**. The reviewer proposed to use the XRD-data of out-of-plane peaks such as (002) and (200)-phase to identify the compound as Mn₂Au. Unfortunately, we cannot prepare (002)-phase Mn₂Au thin films on PMNPT substrate, and the XRD-data of out-of-plane (200)-phase Mn₂Au is very similar to that of (002)-phase MnAu (shown in **Fig. R5**), which can't be used to distinguish whether the material is Mn₂Au. From the FWHM of the (103)-phase Mn₂Au, it indicates that the (103)-phase film is not a perfect quasi-epitaxial orientation film. Although the position of the XRD peak is around 40°, we can explain from three aspects that no doubt our sample is Mn₂Au rather than MnAu. The shift of the diffraction peak is caused by the large lattice mismatch between Mn₂Au and PMN-PT and the resultant large strain in the thin film.

Fig. R4. (a) X-ray diffraction spectrum of Mn₂Au/PMN-PT (011) samples; (b) rocking curve scan of the Mn₂Au (103) peak.

Fig. R5. Comparison of XRD-PDF cards of Mn_2Au and $MnAu$ materials.

i) To carefully check the relative composition ratio between Mn and Au, we take use of Inductively Coupled Plasma (ICP), which is a precise method used for quantitatively identifying and measuring the individual elements in a sample. We measured two samples of (103)-phase Mn_2Au and obtained the results as shown in the **Table 1**. The results show that the composition ratio of Mn and Au is 2:1, which is consistent with the material Mn_2Au , indicating that the sample is not $MnAu$. The typical ICP data of one sample are listed in Table 1

Table1-1: ICP analysis of Mn_2Au .

	Total (ug)	relative atomic mass	Atomic content
Au	18.06	197	33.48%
Mn	9.988	54.94	66.52%

ii) According to the parameters of $MnAu$ ($a = 3.183\text{\AA}$ and $c = 3.283\text{\AA}$, Appl. Phys. Lett. 79, 1507 (2001)), its (101)- or (011)-peaks should locate at 39.2° . In our case, we grow 20, 30, and 50 nm-thick Mn_2Au films, the strain from the substrate would relax gradually with increasing the film thickness. A careful inspection of the XRD peaks of these three samples in **Fig. R6** show that the diffraction peak shifts to a higher angle, from 39.8° to 40.2° , with strain relaxation. If the sample phase were $MnAu$, the diffraction peak of the relaxation status should be more close to 39.2° (a lower angle direction), rather than to the ideal Mn_2Au peak at 41.7° . Thus, the XRD data for the sample with different film thicknesses disclose that the sample phase is Mn_2Au , where the large lattice mismatch between Mn_2Au and PMN-PT results in the strain and comparatively large shift of the diffraction peak.

iii), From the symmetry point of view, the two inversion-symmetry-breaking antiferromagnetic sublattices of Mn_2Au form inversion partners, which makes the material an attractive candidate for Néel spin-orbit torque (Phys. Rev. Lett. 113, 157201 (2014) and Nat. Mater. 18, 931–935 (2019)). However, the MnAu is CsCl-type structure without sublattice-inversion-symmetry-breaking, (Appl. Phys. Lett. 79, 1507 (2001)), precluding the existence of NSOT.

Fig. R6 Enlarged view of X-ray diffraction spectrum from $\text{Mn}_2\text{Au}/\text{PMN-PT}$ (011) samples.

2) Regarding sample characterization, the results of other groups based on completely different Mn_2Au samples cannot be used to draw conclusions regarding the samples discussed here. E.g., the discussion of the antiferromagnetic domain size on page 8 cites work from another group based on epitaxial Mn_2Au (001) samples grown on different substrates. There is no reason to expect similar domains in the completely different samples discussed here. This applies also to the cited work on imaging of current driven domain wall switching on page 8. Please make a clear statement, that no microscopic magnetic characterization of the samples discussed here is available (or is it?).

Reply: The size of the laser spot used in our work is ~ 3 mm, which is much larger than the size of the Néel-type domains of Mn_2Au (Phys. Rev. B 97, 134429 (2018) and Phys. Rev. B 99, 140409 (2019)). On the other hand, the antiferromagnetic domain switching is a multi-domain switching, as revealed by the transport and domain characterization (Science 351, 587–590 (2016), Nat. Mater. 18, 931–935 (2019), and Appl. Phys. Lett. 117, 082401 (2020)). Although it is the (103)-phase Mn_2Au thin film used in our experiment, the Néel-type domain size of the same material could not be wide different. Therefore, we cite these two works as the basis for our experimental interpretation.

In order to eliminate the misunderstanding caused by our description, we revised the parts in the manuscript (Page 8 Line 16–21): **This phenomenon arises because the size of the laser spot used in our work is ~3 mm, which is much larger than the size of the Néel-type domains of Mn₂Au^{31,32}. Especially, XMLD-PEEM directly exhibits that the domain switching is almost 30% with NSOT switching³². The results indicate that the proportion of the Néel vector reorientation in this sample is approximately 30% in the macroscopic region of the sample.**

3) It is beyond doubt that the Femtosecond laser pulse generates THz radiation from the sample. However, the role of the piezoelectric substrate in this process requires further investigations. The laser pulse heats the substrate, which is resulting in anisotropic thermal expansion. This corresponds to the formation of a piezoelectric voltage, which should drive a current in the metal film on top. This could be the origin of THz emission independent from the specific metallic material on top, i.e., could be not related to antiferromagnetism at all.

Please provide reference experiments with a corresponding non-magnetic metal thin film grown on your PNM-PT substrates.

Reply: The control experiment with a corresponding non-magnetic metal thin film Pt (10 nm) grown on PMN-PT (011) substrates shown in **Fig. R7**. No terahertz signal was observed at different sample azimuth angles (0°, 45°, and 90°). It indicates that the emission of terahertz signal comes from the antiferromagnetic material Mn₂Au, rather than from the strain or anisotropic thermal expansion of the piezoelectric substrate.

Fig. R7. THz waveforms from PMN-PT/Pt(10nm) at different sample azimuth angles

(0°, 45°, and 90°).

4) The abstract and introduction suggests, that the existence of a significant Néel spin-orbit torque in Mn₂Au is experimentally established, which is not the case. It should be mentioned that there is strong evidence for dominant thermomagnetoelastic switching in other antiferromagnets (e.g. Nano Lett. 2021, 21, 1, 114–119). Regarding current driven Néel vector switching in Mn₂Au, it is still an open question if NSOT or thermomagnetoelastic mechanisms are stronger. However, regarding potential laser pulse induced heating effects associated with the experiments discussed by the authors, a possible thermomagnetoelastic effect on the Néel vector is highly relevant and should be considered.

Reply: We would like to thank the reviewer for the suggestions on the abstract and introduction, which we have revised in the manuscript. In addition, in this work, we apply electric field to the ferroelectric substrate (PMN-PT) to switch the Néel vector without mentioning the use of laser for the Néel vector switch. However, concerning whether there is a thermo-magnetoelastic effect for Néel vector switching in a whole film sample (which has not been fabricated into a device), our previous work (Appl. Phys. Lett. 119, 212401 (2021)) can also provide an explanation. Based on the simulation data (**Fig. R8**), it is visible that in the same device, the laser heating on sample is negligible, far less than the thermal effect induced by electric current. Therefore, when the laser acts on our thin film sample, the reviewer mentioned that due to the thermo-magnetoelastic effect of laser on the Néel vector switching can be ignored. However, the laser pulse-induced heating effect is able to excite an oscillation of the Néel vector n in a picosecond time scale, resulting in locally non-equilibrium spin polarization with opposite signs on the sublattices.

Fig. R8. Finite-element simulation on eight-terminal Hall cross device (α -Fe₂O₃/Pt). (a) Temperature distribution of current pulse in the simulated device. (b) Temperature

distribution of THz electric pulse in the simulated device.

We have revised the parts in the manuscript (Page 2 Line 4-7): **In addition to the effects associated with spin-orbit torques (SOT) from the heavy metal (HM) layer⁴⁻⁷, the thermo-magnetoelastic effect^{7,8} is also one of the mechanisms for Néel vector switching in the AFM/HM heterostructure.**

We rewrite the sentence in Page 3 Line 5-9: **In this regard, ultrafast optical excitation is suggested as the most promising tool to explore the antiferromagnetic magnetization dynamic¹⁸⁻²³, and is able to excite an oscillation of the Néel vector n in a picosecond time scale²⁰. The charge current generated from the oscillation of n is accompanied by the emission of THz wave.**

5) The authors explain the laser pulse induced THz emission by a “reciprocal” NSOT. Please provide more information about the concept of this mechanism. The introduction describes an SOT generated at the interface between a heavy metal and an antiferromagnet, which is only loosely related to bulk NSOT and therefore misleading. Please explain the NSOT mechanism instead.

Also in the introduction, it is mentioned that a spin current is generated by “antiferromagnetic dynamics”. What is meant here, a spin current carried by magnons?

Please provide information on how the laser pulse couples to antiferromagnetic magnons in a way that a staggered magnetic field is generated at the antiferromagnetic sublattices of Mn₂Au, as is would be required for a (reciprocal) NSOT mechanism.

Reply: We thank the reviewer for the comments. We add the discussion on NSOT in the introduction part of our manuscript (Page 2 Line 17-19): **In AFMs with local inversion asymmetry, such as CuMnAs and Mn₂Au, magnetization switching can be induced without the assistance of HM and achieved only in single AFM film^{13,15,16}.** We have emphasized the generation of NSOT in antiferromagnetic thin film without heavy metal, such as Mn₂Au (Nat Commun 9, 348 (2018), Nat. Mater. 18, 931–935 (2019)), CuMnAs (Science 351, 587–590 (2016)). Besides, we revised the explanation of the NSOT mechanism in Page 2 Line 19-23: **Due to the local inversion symmetry breaking, a current induces non-equilibrium spin polarization. Opposite inversion symmetry breaking in the two sublattices results in staggered spin polarization, leading to the coherent switching of the moments at two**

antiferromagnetic sublattices. Such current-induced torque for Néel vector switching is known as the Néel spin-orbit torque (NSOT)^{13,15-17}.

We deleted the “antiferromagnetic dynamics” and revised the description to avoid misunderstandings in Page 2 Line 7-8: **In turn, the high intrinsic frequency of AFM offers an opportunity to pump ultrafast spin current and generate the ultrafast charge current.**

To illustrate the staggered magnetic field for the reciprocal NSOT, we rewrite the sentence in Page 2 Line 23-25 and Page 3 Line 1-2: **Accordingly, in the reciprocal process of NSOT, magnonic charge current is expected to occur in AFM with local inversion symmetry breaking, when staggered non-equilibrium spin polarization is excited by antiferromagnetic magnetization dynamic, i.e., a fluctuation of the Néel vector n .** And rewrite the sentence in Page 3 Line 5-9: **In this regard, ultrafast optical excitation is suggested as the most promising tool to explore the antiferromagnetic magnetization dynamic¹⁸⁻²³, and is able to excite an oscillation of the Néel vector n in a picosecond time scale²⁰. The charge current generated from the oscillation of n is accompanied by the emission of THz wave.**

We add the emission depends on the polarization of the pump and magnons generation in supporting information S2:

S2 Néel spin-orbit torques and magnonic charge current generation

It is well established that charge current-induced local spin polarization can couple to the local magnetic moment to produce Néel spin-orbit torque (NSOT), which enables a current-induced rotation of the local moments while preserving their Néel vector [1]. Accordingly, in the reciprocal process of NSOT, a charge current is expected to occur in AFM with local inversion symmetry breaking when staggered non-equilibrium spin polarization is excited by antiferromagnetic magnetization dynamic, i.e., a fluctuation of the Néel vector n .

To identify the THz emission mechanisms in Mn₂Au thin films, we measured the time domain THz signals for different linear pump polarization angle α , as shown in Fig. S2. The THz emission signal exhibits independence of the linear pump polarization angle α (**Fig. S2a**), where the azimuth θ of the sample remains the same (see Supporting Information S1 for THz emission measurement setup). Accordingly, **Fig. S2b** shows an isotropic behavior of the THz signal amplitude on the linear pump polarization (angle α), in contrast to a recent report in (111) NiO thin films [2]. This result suggests that thermal

absorption in the metallic material Mn_2Au is the driving mechanism of ultrafast spin-current generation with laser pump pulse irradiating the sample [3].

Fig. S2 (a) Time domain THz emission from Mn_2Au with variation of the linear pump polarization angle α . (b) Dependence of the THz signal on the linear pump polarization (angle α) showing an isotropic behavior. Error bars correspond to standard deviations.

In the crystalline structure of Mn_2Au (**Fig. S3a**), the magnetic sublattices exhibit a locally broken inversion symmetry, and the optical pump pulse irradiates the sample surface. The oscillation of the Néel vector \mathbf{n} is excited by the ultrafast optical pulse, resulting in locally non-equilibrium spin polarization with opposite signs on the sublattices. From the initial state illustrated in **Fig. S3b**, Néel vector \mathbf{n}_0 is in the initial state before the optical pump irradiation, which is represented in purple. When the optical pump is applied, a deflection of the Néel vector $\mathbf{n}(t)$ followed by the oscillation, i.e., magnonic excitation is generated (**Fig. S3c**), due to the thermo-magnetoelastic effect [4] or electron thermal excitation [5] of the metal layer Mn_2Au .

The Holstein-Primakoff transformation for antiferromagnetic magnons indicates spins on site i as $S_{A,i}^n = S_0 - a_i^+ a_i, S_{B,i}^n = -S_0 + b_i^+ b_i$. Here, n labels the equilibrium Néel vector direction, A and B label the sublattices, a_i, a_i^+, b_i, b_i^+ are the annihilation and creation operators for site i at A and B sublattices, respectively, S_0 is the spin of the moment [6]. When magnon is excited, the spins on A and B sublattices derive from $\pm S_0$, result in the local spin polarization ($\sigma_A = -\sigma_B$), which is equal to the deviations $\sigma \mathbf{n}$ of the Néel vector from the equilibrium \mathbf{n}_0 . Thus, a charge current can be described as $\mathbf{J}_c \sim \sigma_{A,B} \times \hat{\mathbf{z}}$ by the reciprocal effect of NSOT, and leads to THz radiation.

Fig. S3. Generation of magnonic charge current. (a) The crystalline structure of Mn_2Au . Magnetic sublattices Mn_A and Mn_B are marked in red and green, respectively. The projection of magnetic sublattices Mn_A and Mn_B on the Mn_2Au (103) plane aligns along the $[\bar{3}31]$ direction, and laser irradiates on the sample surface. (b) The initial state of Néel vector n_0 is represented in purple. (c) Via laser irradiation, a deflection of the Néel vector happens followed by oscillations, i.e. magnon excitations. A magnonic charge current is generated on the Mn_2Au surface by the reciprocal of NSOT, and leads to THz radiation.

Reviewers' Comments:

Reviewer #1:

Remarks to the Author:

Antiferromagnetic magnonic charge current generation via ultrafast optical excitation

In this article, the authors have investigated THz emission from antiferromagnetic material Mn₂Au due to its characteristic broken inversion symmetry.

I thank the author for their detailed response to all my questions. I acknowledge that the authors have satisfactorily answered most of the comments from all the reviewers, and I have gone through each. However, I still have some concerns about some of the explanations, which I point out below.

1. As pointed out by reviewer 1, It is indeed surprising to notice such a low THz pulse generation from Mn₂Au/Pt compared to Mn₂Au. The authors claim that no spin flows in case of Mn₂Au/ Heavy metal bilayer. However, this is a very bold claim. Can the authors explain why generated spin current is not flowing to an adjacent layer?

2. Did the author check the phase of the emitted THz from the bare Mn₂Au before depositing the Pt and W layer on those samples (Fig.S8)? As shown in Fig.S6, THz emission from the same thickness Mn₂Au fabricated simultaneously can have different phases and amplitude, implying the generation of oppositely polarized spin currents. Hence the same polarity in Mn₂Au/Pt and Mn₂Au/W may not be a good indicator to say that the spin current is not flowing to the heavy metal layer. It can be possible that the spin polarization in Mn₂Au/Pt and Mn₂Au/W may be the opposite in the samples that the authors have considered.

Reviewer #2:

Remarks to the Author:

The additional measurements the authors provide in response to the Reviewers' comments help clarifying many aspect of this experiment. In my opinion the main point that remains to be clarified is the mechanism that triggers the effect.

The authors cover this very quickly, suggesting two mechanisms: thermomagnetoelastic effect and electron thermal excitation without justifying their claim. We agree that the starting point should be a non-equilibrium spin on each magnetic sub-lattice $\sigma_A = -\sigma_B$, but how is this non-equilibrium spin generated? The authors keep referring to optically induced oscillations of the Neel vector. If these are not coherent oscillations, as I think, I find Fig.1b of the main text or S3 very misleading. Maybe a better figure would just show a net change in the Neel vector length after optical pumping.

In the case of thermomagnetoelastic switching, what is the temperature gradient and strain we should consider? What do the authors mean exactly by electron thermal excitation? Can't the optical-induced quenching of the magnetisation in metals be sufficient to justify the non-equilibrium spin on each sub-lattice? I suggest the authors consider the work published in PRB 95, 094434 (2017).

Reviewer #3:

Remarks to the Author:

The authors answered some of my questions regarding the first version of the manuscript. However, there are still important clarifications regarding the sample characterization required:

The authors replied that they have re-measured Mn₂Au/PNM-PT samples and now obtained a stronger (103) XRD peak. What was the thickness of the sample used for re-measuring? The same 15 nm as the sample used to obtain the data shown in the manuscript? This would be a

requirement for significant new insights.

Answering my question regarding the width of the XRD-peak and corresponding particle size, the authors concluded a size of 7 nm from their new XRD data. This is only half of the 15 nm thickness, which the sample presumably should have, and indicates a crystallographically inhomogeneous sample. Also, one can see in Fig. R6, that the XRD peak does not become narrower with increasing film thickness, which indicates a granular growth mode. So, it is important to clarify the potential presence of a MnAu impurity phase as described below:

I was asking for an XRD analysis of off-specular XRD peaks, which allow to distinguish the tetragonal Mn₂Au phase from a pseudo-cubic MnAu phase. I was not asking to grow the samples in another orientation. The two phases are similar, but not identical in XRD. Please note that if the 2Theta values of the 2 different phases are similar, this does not necessarily mean that they cannot be distinguished in XRD. As we are dealing with epitaxial thin films (this is assumed by the authors), there is always a characteristic tilt angle (typically Chi in a 4-circle diffractometer) between specular and off-specular peaks, which also distinguishes the 2 phases.

The authors compare in Fig. R5 the calculated powder XRD-spectra of Mn₂Au and MnAu. There, one can see that at about 2Theta = 44° and 78°, only Mn₂Au and no MnAu peaks appear. Please measure one of these peaks by 4-circle XRD using a 15nm sample. This will be clear evidence for the formation of a Mn₂Au phase.

The ICP analysis provides some evidence for a Mn₂Au sample, but could also be misleading, given that the XRD particle size is only 7 nm. What was the thickness of the sample used for the ICP analysis? The MnAu phase could form at the beginning of the growth process, which could make it dominant for thin films.

In their rebuttal letter, the authors use the fact that only in Mn₂Au an NSOT can be obtained as evidence for the existence of a Mn₂Au phase. However, as there is no clear evidence for a NSOT in the samples investigated by the authors, this argument is not valid. On the contrary, exactly because only in Mn₂Au a NSOT is possible, it is absolutely required to prove the existence of this phase.

Regarding the assumptions about the AFM domain size, the authors reply that "the Néel-type domain size of the same material could not be wide different" compared to the results from two references. However, given the small 7 nm XRD-derived grain size of the samples used by the authors (and assuming that the sample is indeed Mn₂Au), it is well possible that their magnetic domains are much smaller than in those of the references. It is true that the laser spot is in any case much larger than the AFM domain size. However, assuming a relation of the AFM domain pattern of the samples from different groups grown in different orientations is misleading and should not be done. At least it should be clearly stated that the samples mentioned for comparison are fully epitaxial and grown in another orientation.

Response Letter of NCOMMS-23-00324A

We have addressed the issues raised by three referees point by point below. The amendments in our revised manuscript are highlighted in red. The main revisions include:

- 1) We added the **analysis of off-specular XRD peaks**, which indicates that the material is Mn_2Au .
- 2) We provided the comparison of Mn_2Au (10 nm), Mn_2Au (10 nm)/Pt (5 nm), Mn_2Au (10 nm) and Mn_2Au (10 nm)/W (5 nm) terahertz time-domain spectrum, we concluded that the **flow of the spin current across the interface is not dominant**.
- 3) We updated **Fig.1b in the manuscript**, where the fluctuation and reduction of magnetic moments coexist.
- 4) We measured the THz signals for the new 15 nm $\text{Mn}_2\text{Au}/\text{PMN-PT}$ sample, and the angular dependence of the amplitude is clearly consistent with the manuscript.
- 5) ICP measurements with different thickness are provided, indicating that the sample is not MnAu .

Response to Reviewer #1:

Antiferromagnetic magnonic charge current generation via ultrafast optical excitation. In this article, the authors have investigated THz emission from antiferromagnetic material Mn_2Au due to its characteristic broken inversion symmetry.

I thank the author for their detailed response to all my questions. I acknowledge that the authors have satisfactorily answered most of the comments from all the reviewers, and I have gone through each. However, I still have some concerns about some of the explanations, which I point out below.

Reply: We are grateful to the reviewer for carefully reviewing our manuscript and positive evaluation “I acknowledge that the authors have satisfactorily answered most of the comments from all the reviewers”

1. As pointed out by reviewer 1, It is indeed surprising to notice such a low THz pulse generation from $\text{Mn}_2\text{Au}/\text{Pt}$ compared to Mn_2Au . The authors claim that no spin flows in case of $\text{Mn}_2\text{Au}/$ Heavy metal bilayer. However, this is a very bold claim. Can the authors explain why generated spin current is not flowing to an adjacent layer?

Reply: In the Mn_2Au thin film, the femtosecond laser pulse excites the non-equilibrium spin polarization with opposite signs on the Mn sublattices.

This staggered spin polarization (*not spin current*) of the electrons is directly converted into a charge current by spin-orbit coupling due to the local inversion symmetry breaking. For *flowing* electron spins, since the spins at the two sublattices are opposite, there is nearly no net spin current. We emphasize that the charge current in Mn₂Au thin films is generated from local spin polarization, instead of spin current.

In the case of Mn₂Au/heavy metal bilayer, if there is a spin current flowing into the heavy metal and the inverse spin Hall effect dominates the THz emission, the intensity of THz signal from Mn₂Au/Pt or Mn₂Au/W can be higher than Mn₂Au. However, Fig. S8 shows lower THz pulse generation from Mn₂Au/heavy metal bilayers compared to Mn₂Au, indicating that the spin current injection is not the dominating origin for THz generation. Instead the non-equilibrium spin polarization in single Mn₂Au layer generates the charge current directly.

2. Did the author check the phase of the emitted THz from the bare Mn₂Au before depositing the Pt and W layer on those samples (Fig.S8)? As shown in Fig.S6, THz emission from the same thickness Mn₂Au fabricated simultaneously can have different phases and amplitude, implying the generation of oppositely polarized spin currents. Hence the same polarity in Mn₂Au/Pt and Mn₂Au/W may not be a good indicator to say that the spin current is not flowing to the heavy metal layer. It can be possible that the spin polarization in Mn₂Au/Pt and Mn₂Au/W may be the opposite in the samples that the authors have considered.

Reply: We agree that the signal polarity is not a good indicator. Here, by analyzing the amplitude of the THz traces from bare and heavy metal-capped Mn₂Au, instead of polarity of the THz traces, we can conclude that the flow of the spin current across the interface is not dominant.

We prepared two 10 nm-thick Mn₂Au samples simultaneously and checked the THz signal from the bare Mn₂Au (the original state) before depositing the Pt and W. As shown in **Fig. R1** that after depositing Pt (**Fig. R1a**) and W (**Fig. R1b**), the intensities of bilayers are both much lower than the bare Mn₂Au. It indicates that the spin current injected in heavy metal is not the dominant reason for the generation of THz emission. Therefore, we believe that Mn₂Au/heavy metal bilayer does not generate spins flowing into heavy metals, and the emission of terahertz signals due to the inverse spin Hall effect at the interface does not contribute.

Fig. R1. Comparison of terahertz time-domain spectrum between Mn₂Au (10 nm) and Mn₂Au (10 nm)/Pt (5 nm) (a), and Mn₂Au (10 nm) and Mn₂Au (10 nm)/W (5 nm) (b).

Response to Reviewer #2:

The additional measurements the authors provide in response to the Reviewers' comments help clarifying many aspect of this experiment. In my opinion the main point that remains to be clarified is the mechanism that triggers the effect.

The authors cover this very quickly, suggesting two mechanisms: thermomagnetoelastic effect and electron thermal excitation without justifying their claim. We agree that the starting point should be a non-equilibrium spin on each magnetic sub-lattice $\sigma_A = -\sigma_B$, but how is this non-equilibrium spin generated? The authors keep referring to optically induced oscillations of the Neel vector. If these are not coherent oscillations, as I think, I find Fig.1b of the main text or S3 very misleading. Maybe a better figure would just show a net change in the Neel vector length after optical pumping.

In the case of thermomagnetoelastic switching, what is the temperature gradient and strain we should consider? What do the authors mean exactly by electron thermal excitation? Can't the optical-induced quenching of the magnetisation in metals be sufficient to justify the non-equilibrium spin on each sub-lattice? I suggest the authors consider the work published in PRB 95, 094434 (2017).

Reply: We are grateful to the reviewer for carefully reviewing our manuscript. We have carefully read the articles (Ref. Phys. Rev. B 95, 094434 (2017)) suggested by the reviewer and the references involved in the articles (Ref. J. Chem. Theory Comput. 11, 4870 (2015) and Ref. Phys. Rev. Lett. 115, 217204 (2015)). This gives us a deeper understanding of the mechanism. We revised Fig.1b in the main text, and the figure schematically shows the coexistence of

two mechanisms. In our experiment, 800 nm, 100 fs laser pump pulse is used to excite THz signal. While short and intense pulse on magnets with weak exchange coupling mainly causes moment reduction, the long excitation time and the strong exchange coupling in Mn_2Au (Ref. Phys. Rev. B 97, 184416 (2018)) allow two processes: fluctuation of the orientations of the local magnetic moments and the magnetization reduction, coexist. Both processes can contribute to the charge current generation (Ref. Phys. Rev. B 95, 094434 (2017)).

The thermal magnetoelastic contribution to the mechanism can be understood as follows. After the heating by the laser pulse, strain is generated due to the different thermal expansion coefficients of the substrate and the Mn_2Au thin film, which drives the fluctuation and reduction of magnetic moments. The temperature gradient is not necessary to be considered since the strain can be directly generated by homogeneous heating and different thermal expansion coefficients.

We revised the description of the Introduction part in Page 3 Line 10 “able to excite a fluctuation of the Néel vector n ”. Page 3 Line 16: “The femtosecond laser pulse triggers the orientation fluctuation^{12, 25, 26} and magnitude reduction²⁷ of the local magnetic moments...”. And Page 3 Line 3 from the bottom “The observed AFM magnonic charge current generated from the magnetic moment fluctuation”. Then, we updated Fig.1b in the manuscript, where the fluctuation and reduction of magnetic moments coexist. We cited the literature recommended by the reviewers as our theoretical basis (Ref. Phys. Rev. B 95, 094434 (2017), Ref. Phys. Rev. Lett. 115, 217204 (2015) and J. Chem. Theory Comput. 11, 4870 (2015)) in Ref. 25, 26, 27.

Fig. 1 Principle of magnonic charge current generation in Mn_2Au . a, The crystalline structure of Mn_2Au . Magnetic sublattices Mn_A and Mn_B are marked in red and green,

respectively. b, The orientation fluctuation, and magnitude reduction of the local magnetic moments in Mn_2Au is triggered by the femtosecond laser pulse, which results in a non-equilibrium spin polarization with the opposite sign on the adjacent Mn sublattices. c, Staggered spin polarization generated at magnetic sublattices, giving rise to the charge current by the reciprocal relationship of NSOT.

In the **Supporting Information S2**, we delete the unclear “electron thermal excitation” and rewrite the description on optical-induced magnetization orientation fluctuation and the quenching of the magnetization. This can be found in **Supporting Information S2** in Page 2 Line 3 from the bottom: “**This result suggests that the ultrafast spin-current generation is driving by isotropic excitations in the metallic Mn_2Au being irradiated by the laser pump pulse [3]**”. And **Supporting Information S2** Page 3 Line 7: “**The fluctuation of the Néel vector n is excited by the ultrafast optical pulse, resulting in locally non-equilibrium spin polarization with opposite signs on the sublattices. From the initial state illustrated in Fig. S3b, Néel vector n_0 is in the initial state before the optical pump irradiation, which is represented in purple. When the optical pump is applied, the Néel vector $n(t)$ experiences magnitude reduction due to the optical quenching of the magnetization [4,5], and the orientation fluctuation in the many-body relaxation process [6] (Fig. S3c).**”

Response to Reviewer #3:

The authors answered some of my questions regarding the first version of the manuscript. However, there are still important clarifications regarding the sample characterization required:

The authors replied that they have re-measured $\text{Mn}_2\text{Au}/\text{PMN-PT}$ samples and now obtained a stronger (103) XRD peak. What was the thickness of the sample used for re-measuring? The same 15 nm as the sample used to obtain the data shown in the manuscript? This would be a requirement for significant new insights.

Reply: We are grateful to the reviewer for carefully reviewing our manuscript. The re-measured XRD data is obtained from the 15-nm thick $\text{Mn}_2\text{Au}/\text{PMN-PT}$, and the new 15-nm thick Mn_2Au shows the same behavior as the data in the manuscript (**Fig. R2**). And we employ ferroelastic strain from the ferroelectric material PMN-PT to switch the Mn_2Au Néel vector, the experimental results of emitted THz signals are summarized in **Fig. R2**, the angular dependence of the amplitude is clearly consistent with the manuscript (Fig. 4d and e).

Fig. R2. The amplitude of THz waveforms as a function of sample azimuth θ with the $+4 \text{ kV cm}^{-1}$ (a) and -2 kV cm^{-1} (b).

Answering my question regarding the width of the XRD-peak and corresponding particle size, the authors concluded a size of 7 nm from their new XRD data. This is only half of the 15 nm thickness, which the sample presumably should have, and indicates a crystallographically inhomogeneous sample. Also, one can see in Fig. R6, that the XRD peak does not become narrower with increasing film thickness, which indicates a granular growth mode. So, it is important to clarify the potential presence of a MnAu impurity phase as described below:

I was asking for an XRD analysis of off-specular XRD peaks, which allow to distinguish the tetragonal Mn_2Au phase from a pseudo-cubic MnAu phase. I was not asking to grow the samples in another orientation. The two phases are similar, but not identical in XRD. Please note that if the 2θ values of the 2 different phases are similar, this does not necessarily mean that they cannot be distinguished in XRD. As we are dealing with epitaxial thin films (this is assumed by the authors), there is always a characteristic tilt angle (typically χ in a 4-circle diffractometer) between specular and off-specular peaks, which also distinguishes the 2 phases.

The authors compare in Fig. R5 the calculated powder XRD-spectra of Mn_2Au and MnAu. There, one can see that at about $2\theta = 44^\circ$ and 78° , only Mn_2Au and no MnAu peaks appear. Please measure one of these peaks by 4-circle XRD using a 15nm sample. This will be clear evidence for the formation of a Mn_2Au phase.

The ICP analysis provides some evidence for a Mn_2Au sample, but could also be misleading, given that the XRD particle size is only 7 nm. What was the thickness of the sample used for the ICP analysis? The MnAu phase could form at the beginning of the growth process, which could make it dominant for thin films.

In their rebuttal letter, the authors use the fact that only in Mn_2Au an NSOT can be obtained as evidence for the existence of a Mn_2Au phase. However, as there is no

clear evidence for a NSOT in the samples investigated by the authors, this argument it not valid. On the contrary, exactly because only in Mn_2Au a NSOT is possible, it is absolutely required to prove the existence of this phase.

Regarding the assumptions about the AFM domain size, the authors reply that “the Néel-type domain size of the same material could not be wide different” compared to the results from two references. However, given the small 7 nm XRD-derived grain size of the samples used by the authors (and assuming that the sample is indeed Mn_2Au), it is well possible that their magnetic domains are much smaller than in those of the references. It is true that the laser spot is in any case much larger than the AFM domain size. However, assuming a relation of the AFM domain pattern of the samples from different groups grown in different orientations is misleading and should not be done. At least it should be clearly states that the samples mentioned for comparison are fully epitaxial and grown in another orientation.

Reply: As mentioned by the referee, the XRD analysis of off-specular XRD peaks would be very useful to distinguish the tetragonal Mn_2Au phase from a pseudo-cubic MnAu phase, because the big differences between lattice constants of the Mn_2Au ($a = 3.328 \text{ \AA}$, and $c = 8.539 \text{ \AA}$) and MnAu (with $a = 3.183 \text{ \AA}$ and $c = 3.283 \text{ \AA}$) phases (Ref. J. Phys. D: Appl. Phys. 48 385001 (2015)). This important reference was added as Ref. 15 in our main text. We measured the film samples (100 nm) with XRD ϕ -scans of off-specular Mn_2Au and MnAu peaks in **Fig. R3**. We need to mention that the signal of 15 nm Mn_2Au film on PMN-PT (110) is too weak for ϕ -scan, so we choose 100 nm sample with a comparatively stronger signal. XRD off-specular ϕ -scan of MnAu (101) was carefully checked according to ICDD-PDF card. There is no diffraction peak in extensive ϕ -scan. A representative XRD off-specular ϕ -scan spectrum from MnAu {011} [equivalent peaks MnAu (101)] is presented in Fig. R3(a). Obviously, we cannot detect any diffraction peaks within $0\text{--}360^\circ$. The situation turns out to be different for the test with the Mn_2Au phase. Figure R3(b) displays XRD off-specular ϕ -scan of Mn_2Au {013} [equivalent peaks of Mn_2Au (103)]. We can see two diffraction peaks from Mn_2Au (013) and Mn_2Au ($0\bar{1}3$) with an angle difference of 105° . Also, there are two superimposed peaks arising from their twin crystal, thus the peaks are marked as Mn_2Au (013)_{a,b} and Mn_2Au ($0\bar{1}3$)_{a,b}. Accordingly, XRD off-specular ϕ -scan of PMN-PT {111} is shown in Fig. R3(c), two diffraction peaks from PMN-PT (111) and ($\bar{1}11$) exist. We depict in **Fig. R4** twin crystal growth modes of our Mn_2Au sample, which should be the source of the antiferromagnetic multi-domain state for the as-grown sample. The XRD spectra reflect that the material grown by magnetron sputtering is Mn_2Au phase rather than MnAu .

Fig. R3. XRD off-specular ϕ -scan of MnAu {011} (a), Mn₂Au {013} (b) and PMN-PT {111} (c).

Fig. R4. (a) Schematic of the PMN-PT (110) substrate, here $a^* = 4.02 \text{ \AA}$, $c^* = 5.69 \text{ \AA}$. (b) Schematic of the tetragonal Mn₂Au (103). Here $a^* = 3.328 \text{ \AA}$, $c^* = 13.13 \text{ \AA}$. (c) The red dashed lines represent the crystal faces of PMN-PT (111) and ($\bar{1}$ 11). (d) The yellow dashed lines represent the crystal faces of Mn₂Au (0 $\bar{1}$ 3)_{a,b}, the green dashed lines represent the crystal faces of Mn₂Au (013)_{a,b}. (e) Schematic of the epitaxy growth modes of Mn₂Au (103) grown on PMN-PT (110) substrate.

In fact, in the previous version, the ICP data are the average value from

two 40 nm Mn₂Au films. Accordingly the referee's suggestion, we re-measured ICP data with thinner samples with the thickness of 15, 20, and 40 nm. It is necessary to mention here that 7 nm film is too thin, and the mass of the film is not enough for the ICP measurements taking the sensitivity into account. The results show that the composition ratio of Mn and Au is quite close to 2:1 in three samples. The composition measurements are strong evidence for the Mn₂Au phase.

Table 1 ICP analysis of Mn₂Au films

	Relative atomic mass(ug)	15 nm	20 nm	40 nm
Mn (mass)	54.94	3.379	4.924	9.953
Atomic Content		63.64%	66.381%	65.42%
Au (mass)	197	6.921	8.942	18.87
Atomic Content		36.36%	33.619%	34.58%

We have proved in **Fig. R3** that the thin film is Mn₂A, so it is reasonable to cite the domain structure and size of the Néel-type domains of Mn₂Au in other references (Phys. Rev. B 97,134429 (2018) and Phys. Rev. B 99, 140409 (2019)). Although it is the (103)-phase Mn₂Au thin film used in our experiment, the Néel-type domain size of the same material could not be wide different.

Reviewers' Comments:

Reviewer #1:

Remarks to the Author:

In this article, the authors have investigated THz emission from antiferromagnetic material Mn₂Au due to its characteristic broken inversion symmetry.

The authors have satisfactorily answered all the comments and the manuscript is highly recommended for publication at Nature Communication.

Reviewer #2:

Remarks to the Author:

I think the authors have clarified several aspects and provide sufficiently convincing evidence. I find the paper interesting and suitable for Nature Communications.

Typo:

Line 123. Should be cancel out -> should cancel out

Reviewer #3:

Remarks to the Author:

Based on the now supplied additional XRD data, some new conclusions regarding the crystallographic sample quality are possible, which effect fundamental claims of the manuscript. This goes beyond my original concern that mainly a spurious antiferromagnetic MnAu phase may be investigated.

However, regarding the XRD based evidence for the real Mn₂Au phase, the authors present scans of the wrong peak. They show a phi scan of off-specular {013}-peaks, which are equivalent to the specular (103)-peak. This adds no additional insights. Based on this, I recommend the rejection of the manuscript, if not any XRD-peak characteristic for Mn₂Au (which is not also to be expected for MnAu) can be presented. The authors write that they could observe the in-plane {013}-peaks (which are the strongest anyway) only based on films with increased thickness. This is related to missing in-plane order, as discussed below. Thus, failing to observe the MnAu{001} in a phi-scan, as shown in Fig. R3a, provides no evidence for the absence of this phase.

If no characteristic Mn₂Au XRD peak can be observed in phi-scans due to missing in-plane order, other types of scans are in principle possible (such as a Theta/2Theta scan at a selected tilt angle Chi if the sample). Anyway, without evidence for the Mn₂Au phase formation, claims regarding physics based on the NSOT, which is very specific for this compound, are not possible.

Even if we assume (without sufficient experimental evidence) that Mn₂Au is investigated, new insight in the limited sample quality are possible: The authors now present the first phi-scan of an off-specular XRD peak of a similar sample (much thicker than the one used in the manuscript). This scan should show strong and sharp peaks for epitaxial thin films, i.e., in a sample with in-plane structural order. However, the broad peaks shown in Fig. R3 are very weak, with an intensity, which is only a factor of three higher than the background. Thus, the samples are NOT epitaxial, i.e., they consist of small crystallites with almost random in-plane orientation. This should be stated clearly in the manuscript and the XRD phi-scan needs to be added to the manuscript to make this obvious.

However, in such samples without in-plane order there is no defined crystallographic easy-axis orientation and all magnetic anisotropy originates from the piezoelectric substrate. This implies that the orientation of the Néel vector, if aligned by straining the substrate, is random with respect to the crystallographic in-plane axis of Mn₂Au. This should be mentioned clearly in the manuscript and it contradicts the assumption (e.g. in section "sample azimuth dependence after Néel vector switching") of switching the Néel vector by piezoelectric strain between specific crystallographic directions.

In summary, for general consideration of this manuscript clear experimental evidence for the

formation of the assumed crystallographic phase needs to be provided. Even if this should be possible, the limited degree of crystallographic order, without in-plane orientation, does not allow for clear conclusions regarding the NSOT mechanism. Thus, in the best case, the manuscript could be published in a more specialized journal.

Reviewer #4:

Remarks to the Author:

I have comments on the structural characterization for their samples.

I partly agree with the comments from Reviewer #3. As Reviewer #3 is concerned, a growth mode and an emerging phase largely depend on the film thickness in general. Particularly, since the lattice mismatch between PMN-PT(110) and Mn₂Au(103) is large, it is natural to suppose that the large lattice mismatch may affect the growth mode and the emerging phase. Thus, the structural characterization should be done using the identical samples which were used for the terahertz emission experiment. The authors did the additional off-specular XRD scan. Although I appreciate their effort, that was done for the sample with "100 nm thickness", which is quite different from the terahertz emission samples with 10 – 15 nm thickness. In that case, the initial growth condition may be hidden and cannot be discussed. I understand the technical difficulty if they employ the thinner sample, but the off-specular XRD scan is not direct evidence that no additional phase appears in a thin layer thickness region. Instead of XRD characterization, for example, I recommend the authors to observe the detailed structure using high-resolution transmission electron microscopy. That could give more direct information about the local crystal structure, grain size, interface quality, film thickness uniformity, and so on. As the authors mentioned in the rebuttal letter, "twin crystal growth modes of our Mn₂Au sample, which should be the source of the antiferromagnetic multi-domain state for the as-grown sample", the size of twin crystal is an important factor to judge whether "the Neel-type domain size of the same material could not be wide different" or not. From this point of view, the detailed structural characterization will be helpful to support the authors' claim.

Related to the above comment on the structural characterization, I have another question for control terahertz emission experiment comparing Mn₂Au (10 nm), Mn₂Au (10 nm)/Pt (5 nm) and Mn₂Au (10 nm)/W (5 nm) shown in S8 of Supporting information. The authors mentioned that the terahertz signal intensity depends on the antiferromagnetic multi-domain state. Were the control experiments done after modulating the antiferromagnetic domain distribution of Mn₂Au by electric field application? How did the authors confirm that all the samples involve similar antiferromagnetic multi-domain states? Strictly speaking, the exactly same antiferromagnetic multi-domain states are required for comparing terahertz signal intensities between them. I am a bit concerned that the capping layer of Pt or W may affect the magnitude of strain in the Mn₂Au layer induced by electric field and result in the different antiferromagnetic multi-domain state. In order to dispel this concern, more discussion about the antiferromagnetic multi-domain state, namely the detailed structural characterization, is again helpful.

Response Letter of NCOMMS-23-00324B

We very much appreciate the positive evaluation of our manuscript (NCOMMS-23-00324B) by Reviewer #1 “The authors have satisfactorily answered all the comments and the manuscript is highly recommended for publication at Nature Communication”. Reviewer #2 “I think the authors have clarified several aspects and provide sufficiently convincing evidence. I find the paper interesting and suitable for Nature Communications.” And we address the issues raised by Reviewer #3 point by point below. Amendments of our revised manuscript are summarized below in bold face style.

The main modifications include:

- 1) We added the XRD phi-scan in the **Supporting Information Note 5**, which indicates that the material is a pure and textured Mn₂Au phase.
- 2) ICP measurements with different thicknesses are provided, **indicating that the sample is not MnAu.**
- 3) The THz amplitude as a function of the azimuth of the new Mn₂Au sample was measured, which is **consistent with the observation in Fig. 4.**

Response to reviewer #3:

1): Based on the now supplied additional XRD data, some new conclusions regarding the crystallographic sample quality are possible, which effect fundamental claims of the manuscript. This goes beyond my original concern that mainly a spurious antiferromagnetic MnAu phase may be investigated.

Reply: We are grateful to the reviewer for raising questions. We admit that Mn₂Au is not an epitaxial single crystal sample but a Mn₂Au(103) textured sample, because the large lattice mismatch between PMN-PT substrate and Mn₂Au film, grown by magnetron sputtering. The reviewer tends to agree with our evidences that our sample shows Mn₂Au phase after two iterations. Again, besides our extensive characterization based on XRD and ICP, from the symmetry point of view, the two inversion-symmetry-breaking antiferromagnetic sublattices of Mn₂Au form inversion partners, which makes the material an attractive candidate for Néel spin-orbit torque (NSOT, Phys. Rev. Lett. 113, 157201 (2014) and Nat. Mater. 18, 931–935 (2019)). That is the crucial factor that determines non-vanished spins can be generated in a Mn₂Au single layer. However, MnAu is CsCl-type structure without sublattice-inversion-symmetry-breaking, (Appl. Phys. Lett. 79, 1507 (2001)), precluding the existence of NSOT.

According to the reviewer’s suggestion, the THz amplitude as a function of

the azimuth of the new Mn₂Au sample (with higher crystal quality) was measured, and presented in Fig. R1. The data are quite consistent to the one shown in Fig. 4d and e. Obviously, only the antiferromagnetic phase Mn₂Au with local symmetry breaking can generate THz signal according to our mechanism proposed in the main text. Such a behavior cannot be observed in the antiferromagnets without local symmetry breaking, such as FeRh (Fig. 3a).

Fig. R1. Amplitude of THz waveforms as a function of sample azimuth θ after applying different electric field of $+4 \text{ kV cm}^{-1}$ (a) and -2 kV cm^{-1} (b).

ICP measurements were carried out in Mn₂Au films with different thicknesses of 15, 20, and 40 nm (7 nm sample is too thin, the weight is too light and easy to be inaccurate). The results show that the composition ratio of Mn and Au is approximately 2:1, confirming that the sample is not MnAu but Mn₂Au.

Table1: ICP analysis of Mn₂Au.

	Relative atomic mass(ug)	15nm	20nm	40nm
Mn	54.94	3.379	4.924	9.953
Atomic Content		63.64%	66.381%	65.42%
Au	197	6.921	8.942	18.87
Atomic Content		36.36%	33.619%	34.58%

2): However, regarding the XRD based evidence for the real Mn₂Au phase, the authors present scans of the wrong peak. They show a phi scan of off-specular {013}-peaks, which are equivalent to the specular (103)-peak. This adds no additional insights. Based on this, I recommend the rejection of the manuscript, if not any XRD-peak characteristic for Mn₂Au (which is not also to be expected for MnAu) can be presented. The authors write that they could observe the in-plane {013}-peaks (which are the strongest anyway) only based on films with increased thickness. This

is related to missing in-plane order, as discussed below. Thus, failing to observe the MnAu{001} in a phi-scan, as shown in Fig. R3a, provides no evidence for the absence of this phase.

If no characteristic Mn₂Au XRD peak can be observed in phi-scans due to missing in-plane order, other types of scans are in principle possible (such as a Theta/2Theta scan at a selected tilt angle Chi if the sample). Anyway, without evidence for the Mn₂Au phase formation, claims regarding physics based on the NSOT, which is very specific for this compound, are not possible.

Reply: Although the XRD peaks of the Mn₂Au (103) and MnAu (101) phases are very close, the XRD analysis of off-specular XRD peaks would be very useful to distinguish the tetragonal Mn₂Au phase from a pseudo-cubic MnAu phase, since the big differences between lattice constants of the Mn₂Au ($a = 3.328 \text{ \AA}$, and $c = 8.539 \text{ \AA}$) and MnAu (with $a = 3.183 \text{ \AA}$ and $c = 3.283 \text{ \AA}$) phases. And the quality of Mn₂Au thin film is indeed not the single crystal, we measured the film samples (100 nm) with XRD ϕ -scans of off-specular Mn₂Au and MnAu peaks in **Fig. R2**. We need to mention here that because the amplitude of 15 nm sample is too weak, we show the data of the thicker sample of 100 nm.

XRD off-specular ϕ -scan of MnAu (101) was carefully checked according to ICDD-PDF card. There is no diffraction peak in extensive ϕ -scan. A representative XRD off-specular ϕ -scan spectrum from MnAu {011} [equivalent peaks MnAu (101)] is presented in Fig. R2(a). Obviously, we cannot detect any diffraction peaks within 0–360°. The situation turns out to be different for the test with the Mn₂Au phase. Figure R2(b) displays XRD off-specular ϕ -scan of Mn₂Au {013} [equivalent peaks of Mn₂Au (103)]. We can see two diffraction peaks from Mn₂Au (013) and Mn₂Au (0 $\bar{1}$ 3) with an angle difference of 105°. Also, there are two superimposed peaks arising from their twin crystal, thus the peaks are marked as Mn₂Au (013)_{a,b} and Mn₂Au (0 $\bar{1}$ 3)_{a,b}. Accordingly, XRD off-specular ϕ -scan of PMN-PT {111} is shown in Fig. R2(c), two diffraction peaks from PMN-PT (111) and ($\bar{1}$ 11) exist. We depict in **Fig. R3** twin crystal growth modes of our Mn₂Au sample, which should be the source of the antiferromagnetic multi-domain state for the as-grown sample. The XRD spectra reflect that the material grown by magnetron sputtering is Mn₂Au phase rather than MnAu.

Fig. R2. XRD off-specular ϕ -scan of MnAu {011} (a), Mn₂Au {013} (b) and PMN-PT {111} (c).

Fig. R3. (a) Schematic of the PMN-PT (110) substrate, here $a^* = 4.02 \text{ \AA}$, $c^* = 5.69 \text{ \AA}$. (b) Schematic of the tetragonal Mn₂Au (103). Here $a^* = 3.328 \text{ \AA}$, $c^* = 13.13 \text{ \AA}$. (c) The red dashed lines represent the crystal faces of PMN-PT (111) and $(\bar{1}\bar{1}1)$. (d) The yellow dashed lines represent the crystal faces of Mn₂Au $(0\bar{1}3)_{a,b}$, the green dashed lines represent the crystal faces of Mn₂Au $(013)_{a,b}$. (e) Schematic of the epitaxy growth modes of Mn₂Au (103) grown on PMN-PT (110) substrate.

3): Even if we assume (without sufficient experimental evidence) that Mn₂Au is investigated, new insight in the limited sample quality are possible: The authors now

present the first phi-scan of an off-specular XRD peak of a similar sample (much thicker than the one used in the manuscript). This scan should show strong and sharp peaks for epitaxial thin films, i.e., in a sample with in-plane structural order. However, the broad peaks shown in Fig. R3 are very weak, with an intensity, which is only a factor of three higher than the background. Thus, the samples are NOT epitaxial, i.e., they consist of small crystallites with almost random in-plane orientation. This should be stated clearly in the manuscript and the XRD phi-scan needs to be added to the manuscript to make this obvious.

However, in such samples without in-plane order there is no defined crystallographic easy-axis orientation and all magnetic anisotropy originates from the piezoelectric substrate. This implies that the orientation of the Néel vector, if aligned by straining the substrate, is random with respect to the crystallographic in-plane axis of Mn₂Au. This should be mentioned clearly in the manuscript and it contradicts the assumption (e.g. in section “sample azimuth dependence after Néel vector switching”) of switching the Néel vector by piezoelectric strain between specific crystallographic directions.

In summary, for general consideration of this manuscript clear experimental evidence for the formation of the assumed crystallographic phase needs to be provided. Even if this should be possible, the limited degree of crystallographic order, without in-plane orientation, does not allow for clear conclusions regarding the NSOT mechanism. Thus, in the best case, the manuscript could be published in a more specialized journal.

Reply: We add the description of XRD data in the manuscript in Methods (Page 10 Line 18-20): **X-ray diffraction of the Mn₂Au films were measured using Cu K α ₁ radiation with $\lambda = 1.5406 \text{ \AA}$, showing Mn₂Au(103) texture (See Supplementary Note 5).**

Although our material is not a single crystal sample, the sample has a certain orientation. Very strong evidence of the existence of NOST has been obtained in Ref. Nat. Mater. 18, 931–935 (2019), Which also proves that it is a physical phenomenon unique to Mn₂Au characteristics. We agree with the Néel vector of Mn₂Au is aligned by straining of the substrate, however, it does not contradict the assumption of switching the Néel vector by piezoelectric strain between specific crystallographic directions, because here the crystallographic directions are for the PMN-PT substrates. Notably, the antiferromagnetic Néel order switching is not entirely dependent on the crystalline quality (Nat. Commun. 12, 3828 (2021)), the magnetic switching also can be observed in polycrystalline phase of antiferromagnetic materials. Therefore, although the Mn₂Au film is not epitaxial, there is a certain (103)

orientation, they consist of small crystallites with almost (103) orientation.

The sample growth and characterization of Mn_2Au has also been explained in the manuscript in Supporting Information Note 5:

Fig. S7 shows the x-ray diffraction (XRD) spectrum of 15 nm-thick Mn_2Au film grown on PMN-PT (110) substrate. Besides of the diffraction peaks from PMN-PT (011) substrate, only Mn_2Au (103) peak is visible, indicating the growth of (103)-oriented Mn_2Au films. We measured the film samples (100 nm) with XRD ϕ -scans of off-specular Mn_2Au peaks in Fig. S8. Fig. S8(a) shows an XRD ϕ -scan of off-specular of Mn_2Au (103) and equivalent peaks of Mn_2Au (013) and Fig. S8 (b) shows an XRD ϕ -scan of off-specular of PMN-PT (110) and equivalent peaks of PMN-PT (111). It indicates two equivalent growth modes of this sample, reflecting the antiferromagnetic multi-domain state for the as-grown sample. This diffraction peak indicates that the material grown by magnetron sputtering is a pure and oriented Mn_2Au phase.

Fig. S7. X-ray diffraction spectrum of $\text{Mn}_2\text{Au}/\text{PMN-PT}$ (110) samples.

Fig. S8. (a) XRD ϕ -scans of off-specular 100 nm Mn_2Au (013) films. (b) XRD ϕ -scan of off-specular PMN-PT (111) substrate.

Reviewers' Comments:

Reviewer #4:

Remarks to the Author:

I appreciate that the authors added the careful structural analyses involving the transmission electron microscope observation and addressed almost all the concerns I had at the previous round. Although I am mostly satisfied with the authors replies and revisions, unfortunately their answer has raised new concern. Before I recommend the publication, the following problem should be addressed.

In their response letter, the authors wrote "We used a comparatively large substrate ($5 \times 10 \text{ mm}^2$) to prepare Mn₂Au (10 nm) single layer. After terahertz emission measurement from the Mn₂Au single layer, the sample was cut into two identical size ($5 \times 5 \text{ mm}^2$) and covered with Pt (5 nm) and metal W (5 nm) respectively". How did the authors get the very clean interface between Mn₂Au and the heavy metal layer even after the terahertz emission measurement? Ion-cleaning prior to deposition although that might vary the quality of Mn₂Au? In-situ terahertz emission experiment? In general, a high-quality interface is indispensable for good spin current transmission. So, many reliable experiments have been done for bilayer films grown without breaking the vacuum condition to keep the clear interface. Compared to the case using YIG (that is an oxide), Mn₂Au is not so strong against the surface oxidation. The oxidized interface with contaminations may block the spin current transmission from the Mn₂Au to the heavy metal layer. Considering this point, my question is "the control experiment using Mn₂Au / Pt and Mn₂Au / W is really a control experiment to eliminate the possibility of spin current flowing?" If the interface does not have good spin current transmission property, the authors can NOT say "we believe that Mn₂Au/heavy metal bilayer does not generate spins flowing into heavy metals, and the emission of terahertz signals due to the inverse spin Hall effect at the interface does not contribute".

Response Letter of NCOMMS-23-00324D

We very much appreciate the positive evaluation of our manuscript (NCOMMS-23-00324D) by Reviewer #4 “I am mostly satisfied with the authors replies and revisions...” And we address the issues raised by Reviewer 4# point by point below. Our manuscript is revised below in bold face style. We revised the description of the control experiment in **Supporting Information Note 3**.

Response to reviewer #4:

I appreciate that the authors added the careful structural analyses involving the transmission electron microscope observation and addressed almost all the concerns I had at the previous round. Although I am mostly satisfied with the authors replies and revisions, unfortunately their answer has raised new concern. Before I recommend the publication, the following problem should be addressed.

In their response letter, the authors wrote “We used a comparatively large substrate (5×10 mm²) to prepare Mn₂Au (10 nm) single layer. After terahertz emission measurement from the Mn₂Au single layer, the sample was cut into two identical size (5 × 5 mm²) and covered with Pt (5 nm) and metal W (5 nm) respectively”. How did the authors get the very clean interface between Mn₂Au and the heavy metal layer even after the terahertz emission measurement? Ion-cleaning prior to deposition although that might vary the quality of Mn₂Au? In-situ terahertz emission experiment? In general, a high-quality interface is indispensable for good spin current transmission. So, many reliable experiments have been done for bilayer films grown without breaking the vacuum condition to keep the clear interface. Compared to the case using YIG (that is an oxide), Mn₂Au is not so strong against the surface oxidation. The oxidized interface with contaminations may block the spin current transmission from the Mn₂Au to the heavy metal layer. Considering this point, my question is “the control experiment using Mn₂Au / Pt and Mn₂Au / W is really a control experiment to eliminate the possibility of spin current flowing?” If the interface does not have good spin current transmission property, the authors can NOT say “we believe that Mn₂Au/heavy metal bilayer does not generate spins flowing into heavy metals, and the emission of terahertz signals due to the inverse spin Hall effect at the interface does not contribute”.

Reply: We do appreciate that the referee gave us helpful suggestions. We did not perform the in-situ terahertz emission experiment and the interface with contamination may block the spin current transmission from the Mn₂Au to the heavy metal layer. We utilized two sample preparation methods to create control samples. One method to prevent contamination of the interface is to

prepare three samples in situ: Mn₂Au (10 nm), Mn₂Au (10 nm)/Pt (5 nm), and Mn₂Au (10 nm)/W (5 nm). This allows for the transmission of spin currents from the Mn₂Au layer to the heavy metal layer (**Method 1**). The data of THz emission are shown in **Fig. R1(a)**. Mn₂Au single layer shows the strongest amplitude of THz signal and the polarity of THz signals does not reverse with capping layer W and Pt. For **Method 2**, we prepared a 10 nm-thick Mn₂Au single layer on a relatively large substrate (5 × 10 mm²). After THz emission measurement from the Mn₂Au single layer, the sample was divided into two identical pieces (5 × 5 mm²) and coated with Pt (5 nm) and metal W (5 nm) respectively, which ensure that the comparison samples are of the comparable antiferromagnetic multi-domain state. The amplitude of Terahertz signal from Mn₂Au single layer, Mn₂Au/Pt and Mn₂Au/W is compared in **Fig. R1(b)**. The heavy metals Pt and W have opposite spin Hall angles. However, the polarity of terahertz emission does not reverse, and this phenomenon is very similar to the results for **Method 1**.

Although both of these control experimental methods have some drawbacks, they yield similar results, indicating that capping the heavy metal layer does not improve the THz signal of the Mn₂Au single layer. These results suggest that the strong THz signal emitted by the Mn₂Au single layer is due to the spin charge conversion within the film itself. Whether the spin current is transmitted through the interface layer to enhance the THz signal does not impact our understanding of the high strength of the THz signal emitted by the Mn₂Au single layer.

We revised the description of the control experiment “we believe that Mn₂Au/heavy metal bilayer does not generate spins flowing into heavy metals, and the emission of terahertz signals due to the inverse spin Hall effect at the interface does not contribute” in Supporting Information Note 3: **Therefore, Mn₂Au/heavy metal bilayer does not show an enhancement of THz signal compared to Mn₂Au single layer.**

Fig. R1 Terahertz time-domain spectrum of Mn₂Au (10 nm), Mn₂Au (10 nm)/Pt (5 nm) and Mn₂Au (10 nm)/W (5 nm) with Method 1 (a) and Method 2 (b).

Reviewers' Comments:

Reviewer #4:

Remarks to the Author:

The authors showed the additional information about the control experiments, which supports their claim, "Therefore, Mn₂Au/heavy metal bilayer does not show an enhancement of THz signal compared to Mn₂Au single layer." So, I am satisfied with the authors replies.

However, the revision the authors made was minimal. The authors just changed a few sentences. I think that both experiments done by "Method 1" and "Method 2" involve important knowledge and are helpful for readers to understand the difficulty of the control experiment. Thus, the authors should show both control experiments done by "Method 1" and "Method 2" in their Supporting Information No. 3 and explain the details of control experiment as they wrote in their Response Letter.

Response Letter of NCOMMS-23-00324E

We very much appreciate for accepting our manuscript (NCOMMS-23-00324E). And we revised the description of the control experiment in **Supporting Information Note 3** raised by Reviewer 4#.

Response to reviewer #4:

The authors showed the additional information about the control experiments, which supports their claim, "Therefore, Mn₂Au/heavy metal bilayer does not show an enhancement of THz signal compared to Mn₂Au single layer." So, I am satisfied with the authors replies.

However, the revision the authors made was minimal. The authors just changed a few sentences. I think that both experiments done by "Method 1" and "Method 2" involve important knowledge and are helpful for readers to understand the difficulty of the control experiment. Thus, the authors should show both control experiments done by "Method 1" and "Method 2" in their Supporting Information No. 3 and explain the details of control experiment as they wrote in their Response Letter.

Reply: We do appreciate that the referee gave us helpful suggestions and their recognition of our responses. We revised the description of the control experiment and explain the details of the control experiments done by "Method 1" and "Method 2" in **Supporting Information Note 3**:

Note 3. Effect of heavy metals on terahertz emission

We utilized two sample preparation methods to create control samples. One method to prevent contamination of the interface is to prepare three samples in situ: Mn₂Au (10 nm), Mn₂Au (10 nm)/Pt (5 nm), and Mn₂Au (10 nm)/W (5 nm). This allows for the transmission of spin currents from the Mn₂Au layer to the heavy metal layer (Method 1). The control THz emission experiments are performed in Fig. S4a, Mn₂Au single layer shows the strongest amplitude of THz signal and the polarity of THz signals does not reverse with capping layer W and Pt. Method 2 involves preparing a single layer of Mn₂Au (10 nm) on a relatively large substrate

($5 \times 10 \text{ mm}^2$). After THz emission measurement from the Mn_2Au single layer, the sample was divided into two identical pieces ($5 \times 5 \text{ mm}^2$) and coated with Pt (5 nm) and metal W (5 nm) respectively, which ensure that the comparison samples are of the comparable antiferromagnetic multi-domain state. The amplitude of Terahertz signal from Mn_2Au single layer, $\text{Mn}_2\text{Au}/\text{Pt}$ and $\text{Mn}_2\text{Au}/\text{W}$ is compared in Fig. S4b. The heavy metals Pt and W have opposite spin Hall angles. However, the polarity of terahertz emission does not reverse, and this phenomenon is very similar to the results of previous experiments with the Method 1.

They yield similar results, indicating that capping the heavy metal layer does not improve the THz signal of the Mn_2Au single layer. These results suggest that the strong THz signal emitted by the Mn_2Au single layer is due to the spin charge conversion within the film itself. Whether the spin current is transmitted through the interface layer to enhance the THz signal does not impact our understanding of the high strength of the THz signal emitted by the Mn_2Au single layer.

Fig. S4 Terahertz time-domain spectrum of Mn_2Au (10 nm), Mn_2Au (10 nm)/Pt (5 nm) and Mn_2Au (10 nm)/W (5 nm) with Method 1 (a) and Method 2 (b).